# Bias Assessment and Data Drift Detection in Medical Image Analysis: A Survey

## Abstract

Machine learning (ML) models have achieved expert-level performance across a range of diagnostic tasks in medical image analysis, yet their adoption in clinical practice remains limited due to concerns over reliability, fairness, and robustness. Two key threats to trustworthy deployment are bias, arising primarily during model development, and data drift, which occurs post-deployment as data distributions change over time. Although conceptually distinct, these two phenomena are often conflated in the literature or addressed in isolation, despite their potential to interact and jointly undermine model performance. We argue that clearly distinguishing between bias and data drift is essential for developing appropriate reliability strategies: methods designed to mitigate bias during training differ fundamentally from those needed to detect and manage drift in deployment. In this survey, we therefore bring these perspectives together within a unified framework, clarifying their boundaries while also highlighting where they intersect. We present a comprehensive review of methods for assessing and monitoring ML reliability in medical image analysis, focusing on disease classification models. We first define and distinguish bias and data drift, illustrate their manifestations in clinical contexts, and categorise their sources. We then review state-of-the-art approaches for bias encoding assessment and data drift detection, as well as methods for estimating model performance degradation when ground truth labels are not immediately available. Our synthesis highlights methodological gaps, particularly in evaluating drift detection techniques on real-world medical data, and outlines open challenges for future research. By consolidating these perspectives and providing accessible explanations for both technical and clinical audiences, this work aims to support collaboration between developers, clinicians, and healthcare institutions in building fair, transparent, and reliable ML systems for clinical use.

## 1 Introduction

In recent years, technological and algorithmic advances have enabled machine learning (ML) models to achieve expert-level performance in a range of diagnostic and prognostic tasks across medical imaging modalities (Banerjee et al., 2023). From identifying tumors in radiological scans to predicting cardiovascular risks in ultrasound imaging, these models have shown considerable promise as clinical decision support tools (Jones et al., 2023). However, despite this potential, their integration into routine clinical practice remains limited (Salahuddin et al., 2022). A key barrier to adoption is the lack of robustness and trustworthiness across diverse patient populations and clinical environments required for real-world deployment in safety-critical healthcare environments (Saw & Ng, 2022).

Two central challenges to achieving reliable ML in medical image analysis are bias and data drift. Bias typically arises during model development when training data fail to capture the heterogeneity of the target population. For example, models trained predominantly on adult imaging data may underperform on pediatric cases due to anatomical and physiological differences. Additionally, systemic disparities in healthcare access can result in the underrepresentation of certain demographic groups in training sets (DeBenedectis et al., 2022), leading to a degraded model performance for these groups (Seyyed-Kalantari et al., 2020; 2021).

Such biases risk perpetuating or exacerbating existing healthcare inequities (Obermeyer et al., 2019; Char et al., 2018; Gianfrancesco et al., 2018; Glocker et al., 2023a).

In contrast, data drift refers to changes in the input data distribution that occur between model training and deployment (Sahiner et al., 2023). These shifts may be abrupt or gradual, and can stem from variations in imaging hardware, acquisition protocols, patient demographics, or clinical workflows (Kore et al., 2024). For instance, a model developed using data from one hospital may fail when applied in another setting with different equipment or population characteristics. Even within a single hospital, shifts in patient composition (*e.g.* during a pandemic) or the introduction of new scanners can lead to data drift, degrading model performance in ways that often go undetected until clinical harm occurs (Duckworth et al., 2021).

Although conceptually and operationally distinct, bias and data drift are closely linked through their shared impact on model reliability, a prerequisite for safe clinical deployment. Nevertheless, they are rarely addressed together in the literature and are frequently conflated in terminology and scope. Bias arises from static conditions in data curation and model development (pre-deployment), whereas data drift manifests dynamically in deployed systems facing evolving clinical environments (post-deployment). Consequently, mitigation strategies differ: bias requires fair data collection and algorithmic fairness methods, while data drift necessitates ongoing monitoring and model recalibration or retraining (Kore et al., 2024).

Despite their critical importance, no unified framework currently delineates bias and data drift in a systematic way. Existing studies often address them interchangeably, without making their distinction explicit (Drukker et al., 2023a; Vrudhula et al., 2024; Tejani et al., 2024a; Koçak et al., 2025; Hasanzadeh et al., 2025). Other works focus narrowly on one dimension, either data drift (Gama et al., 2014; Kore et al., 2024) or bias (Brown, 2017; Puyol-Antón et al., 2021; Stanley et al., 2024a), or even just on a single facet of one of them (Omar et al., 2025; Godau et al., 2025; Abdullahi et al., 2025). Yet considering bias and data drift in isolation is inadequate for developing reliable ML systems, as the two can interact in subtle but consequential ways. For instance, drift within already underrepresented subgroups may remain undetected when performance is assessed only at the population level, allowing existing biases to mask the emergence of data drift (Khoshravan Azar et al., 2023). As shown in Figure 1, these phenomena emerge at different stages of the ML lifecycle but jointly determine a model's reliability in real-world deployment.

Motivated by these challenges, this survey brings together both topics, bias and data drift, in a unified review focused on reliability in medical image analysis. The reviewed works were gathered through targeted keyword searches and screening of recent publications in major medical imaging and machine learning venues. Unlike existing surveys that treat these concepts independently, our review links pre-deployment bias assessment with post-deployment drift monitoring and performance estimation, providing a lifecycle perspective tailored to clinical deployment. By addressing both phenomena side-by-side, we aim to clarify their definitions, differentiate their causes and implications, and survey existing methodologies for detecting and mitigating their effects. Moreover, our review is positioned to bridge the gap between technical and clinical communities: we intentionally use accessible language and illustrative examples to make these concepts understandable to readers with limited background in computer science or machine learning. This is essential in an interdisciplinary field where close collaboration between developers, clinicians, but also research and healthcare institutions is vital.

**The key contributions of this work are as follows:**

- We establish precise and operational definitions of *bias* and *data drift*, clarifying their distinct roles in the context of machine learning (ML) reliability for medical image analysis.
- Concepts are presented in accessible language to engage both technical and clinical audiences, ~~minimizing~~ minimising reliance on domain-specific jargon.
- Illustrative examples from the medical domain demonstrate how bias and data drift manifest in real-world clinical settings.
- A structured taxonomy is proposed to ~~categorize~~ categorise sources of bias and data drift, including patient-, device-, and workflow-level factors.
- We review current methods for detecting and mitigating bias in medical image analysis, with particular focus on fairness and representational adequacy.

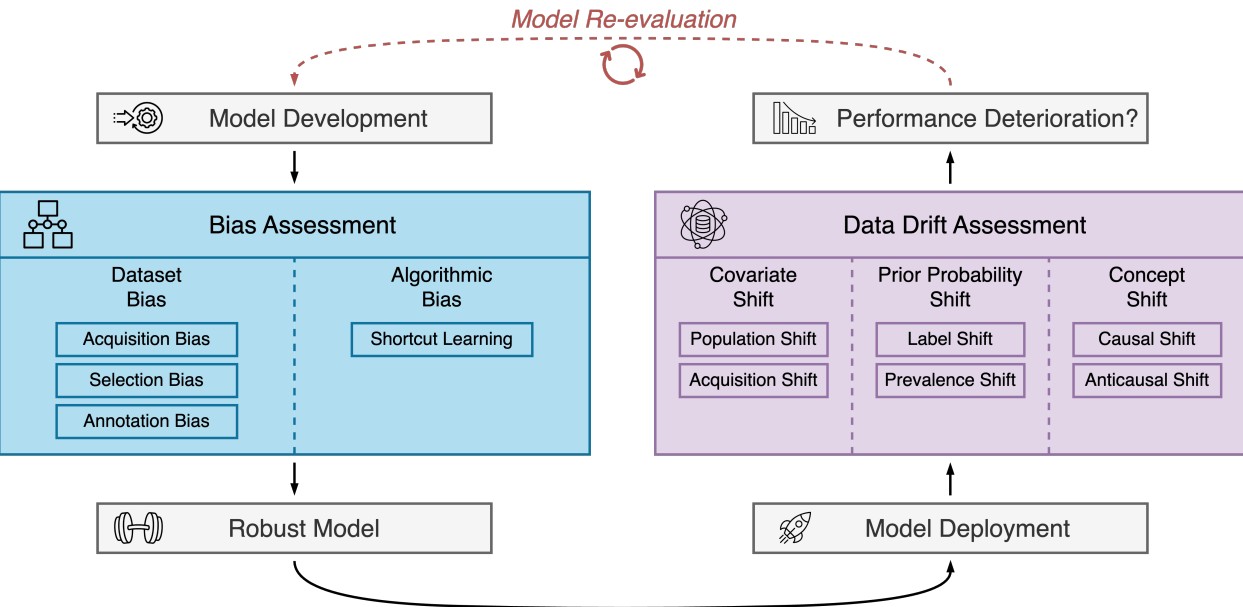

Figure 1: Reliability assessment lifecycle for machine learning models in medical image analysis, showing that bias assessment occurs during model development and data drift assessment during deployment when the model faces evolving clinical environments. Bias is ~~categorized~~ categorised into dataset bias, including acquisition, selection and annotation bias, and algorithmic bias, such as shortcut learning. Data drift is ~~categorized~~ categorised into covariate shift (population shift, acquisition shift), prior probability shift (label shift, prevalence shift) and concept shift (causal shift, anticausal shift). This taxonomy is used throughout the paper to distinguish bias and data drift and to highlight their distinct but complementary roles in ensuring robustness, fairness and ~~long term~~ long-term reliability of clinical ML systems.

- State-of-the-art techniques for identifying and quantifying data drift are surveyed, emphasizing their relevance for maintaining clinical model performance.
- We highlight key challenges and open research questions, outlining opportunities to improve the fairness, robustness, and long-term reliability of clinical ML systems.

The remainder of this ~~paper is organized as follows. Section~~ survey is structured to move from foundational concepts to practical assessment methods and an integrated discussion of reliability in clinical deployment.

Section 2 introduces the key concepts of bias, ~~including its implications , types, and associated assessment metrics and challenges . Section 3 presents the main concepts~~. We describe the different forms of bias that may arise such as selection bias, annotation bias, and algorithmic bias and discuss their implications for model performance and fairness. The section also reviews commonly used fairness metrics, explains when they are appropriate, and highlights challenges such as metric incompatibility, subgroup definition, and clinically motivated prevalence differences.

Section 3 provides an overview of data drift ~~, its clinical relevance , and assessment approaches, along with the inherent challenges in evaluation. Sections~~ and its relevance to clinical workflows. We distinguish between prevalence shift, covariate shift, and concept shift; illustrate how these occur in real medical imaging pipelines; and summarise assessment strategies based on prediction changes, confidence and calibration measures, and distributional statistics. The section also outlines the practical difficulties of drift evaluation, including the scarcity of ground truth and the challenges of determining clinically meaningful shifts.

Sections 4 and 5 review state-of-the-art methods for ~~bias encoding assessment and data drift detection, respectively. Finally, Section 6 provides an integrated discussion of findings, and Section~~ detecting encoded bias and identifying data drift in practice. For bias encoding, we cover approaches that analyse internal features, probe model representations, or isolate the contribution of sensitive and non-sensitive attributes.

For drift detection, we summarise classifier-based, feature-based, and metadata-based techniques, alongside dimensionality-reduction and uncertainty-based approaches, and include representative clinical examples where available.

Section 6 synthesises these components into an integrated reliability perspective, discussing how bias and drift interact, the limitations of current methods, and open challenges for deploying fair and stable models in clinical settings.

Section 7 concludes with ~~open research challenges and future directions~~ a short summary of key insights and remaining research needs for lifelong monitoring and governance of medical image analysis systems.

## 2 Key Concepts of Bias

Bias refers to an estimate of a statistic being systematically different from its population value. If estimates were unbiased on the population level, models would generalise well to other datasets (Wachinger et al., 2019). In the context of ML, ~~we follow~~ recent literature (Seyyed-Kalantari et al., 2020; Brown et al., 2023) ~~and define~~ defines bias as performance disparities across defined subgroups, such as those distinguished by race, age, or gender (Cheong et al., 2023). These attributes are commonly referred to as *sensitive attributes*, as they are associated with socially or clinically meaningful categories that should not unjustifiably affect algorithmic outcomes. However, as noted by Norori et al. (2021), technical or structural biases, such as those arising from data quality, acquisition protocols, or site-specific practices, can reinforce or be misinterpreted as demographic biases if they are correlated with patient characteristics. Similarly, Koçak et al. (2024) emphasise that such non-demographic confounders can systematically affect model behaviour, even in the absence of explicit demographic information. We therefore consider bias in a broader sense, encompassing both demographic and non-demographic sources of systematic error, while maintaining a specific focus on demographic bias due to its direct implications for health equity.

Understanding the causal structure between features and outcomes is essential for identifying and addressing bias. In a *causal* relationship, the input variable $X$ (*e.g.* an image) influences the label $Y$ (*e.g.* diagnosis); this is the typical setting assumed in most supervised learning tasks. In contrast, an *anticausal* relationship occurs when $Y$ is the true causal driver of $X$, such as when disease status causes observable changes in imaging. Many medical tasks are inherently anticausal, which complicates bias detection and mitigation, as correlations between sensitive attributes and $X$ or $Y$ may reflect underlying structural or social confounding rather than spurious noise.

Although identifying the source of bias is essential for selecting the appropriate bias mitigation strategies (Cheong et al., 2023), it is a non-trivial task. Contributing to this difficulty are the complexity of high-dimensional statistical functions (*e.g.* ML models), the entanglement of multiple sources of bias (Koçak et al., 2024), and the presence of unobservable confounding factors in the data (Cheong et al., 2023). Słowik & Bottou (2021) outline two sides to the bias debate: one focuses on *data*, and the other one focuses on *algorithms*. We adhere to this distinction, and in the remainder of this section (*cf.* Sections 2.1–2.2), we contextualise these challenges specifically within the medical domain.

### 2.1 Dataset Bias

A dataset can be conceptualised as a finite, and potentially biased, sample drawn from the true joint probability distribution of the real world, denoted as $P_{REAL}(X, Y)$, where $P_{REAL}(X)$ represents the distribution over input samples (*e.g.* medical images), and $P_{REAL}(Y)$ represents the distribution over corresponding labels (*e.g.* diagnostic annotations). Ideally, a dataset is considered unbiased if its empirical joint distribution $P_{MODEL}(X, Y)$ closely mirrors $P_{REAL}(X, Y)$. However, in practice, sampling is influenced by selection mechanisms, *e.g.* of clinical, demographic, or technical nature, that determine which instances are included in the dataset. These mechanisms can lead to deviations in $P_{MODEL}(X)$ from $P_{REAL}(X)$. Furthermore, when annotations are involved, the process of assigning labels, whether through human judgment or algorithmic heuristics, can distort the conditional distribution $P_{MODEL}(Y|X)$. This introduces annotation bias, whereby the observed labels systematically deviate from the true labels, particularly if certain subpopulations are mislabelled more frequently (Cheong et al., 2023).

Consequently, bias in the dataset, reflected in the misalignment between $P_{MODEL}(X,Y)$ and $P_{REAL}(X,Y)$, can arise from two main sources: the input distribution $X$, due to selective sampling, and the label distribution $Y$, due to annotation errors. Thus, the resulting dataset encodes not only information about the real world, but also the imperfections and asymmetries of the processes used to collect and label the data.

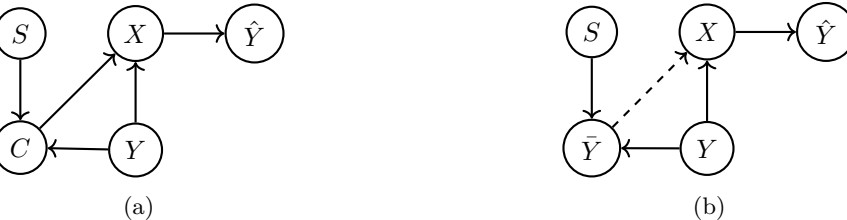

(a)                                         (b)

Figure 2: Graphical models illustrating (a) Selection Bias: A selection mechanism $C$ (*e.g.* which gets included in the dataset) is influenced by both sensitive attribute $S$ and label $Y$, which then influences the input $X$, and (b) Annotation Bias: The observed label $\bar{Y}$ is affected by both the true label $Y$ and the sensitive attribute $S$, introducing systematic errors into the ground truth. The error in the observed label $\bar{Y}$ can influence the selection of input $X$.

### 2.1.1   Acquisition bias

Acquisition bias is a persistent challenge in medical imaging, arising from systematic variability in how images are captured across institutions, or even within the same healthcare facility over time. This form of bias affects the input distribution $P_{MODEL}(X)$ and can undermine downstream model generalisability when acquisition characteristics are not adequately represented or harmonised.

A primary source of acquisition bias is domain shift (Guan & Liu, 2021), which arises from differences in scanner hardware and imaging protocols. For instance, T1-weighted MRI scans acquired from the same patient can exhibit notable differences in contrast and brightness when obtained using scanners from different manufacturers, models, or deployment sites (Opfer et al., 2023). Similarly, in digital pathology, variability in staining protocols, scanner types, compression methods, and optical settings introduces substantial distribution shifts (Madabhushi & Lee, 2016; Dimitriou et al., 2019). To this extent, these acquisition-related biases should be documented using standardised tools such as datasheets for datasets (Mbakwe et al., 2023). Although acquisition differences may appear purely technical, they can lead to structural confounding when specific patient groups are consistently imaged using particular scanners or protocols associated with certain hospitals or regions.

### 2.1.2   Selection Bias

Dataset bias regarding the input $X$ can result from selection bias (Wachinger et al., 2019), see Figure 2(a). The selection mechanism $C$ of samples can be influenced by a sensitive attribute and the label $Y$. This can arise when the participants included in the study do not accurately represent the overall population (Wachinger et al., 2019), *e.g.* when a specific subgroup is over- or under-represented compared to others (Banerjee et al., 2023). Selection bias in clinical practice is evident when, *e.g.* radiologic images are frequently collected from just one or a few locations, resulting in a lack of geographic and racial diversity. Additionally, systemic disparities can lead to variations in image quality, with *e.g.* Black and Hispanic patients sometimes receiving lower quality and less advanced imaging for similar symptoms in emergency departments, especially in patient cost-driven settings like the US (DeBenedectis et al., 2022).

Selection bias can lead to severe class imbalance causing ML models to predominantly learn from the majority class. As a result, these models tend to achieve high performance metrics for the that class, but fail to generalise effectively to any minority classes (Banerjee et al., 2023). Similar challenges occur with demographic factors like gender and socio-economic status, where ML models tend to perform more effectively for the demographic groups that are disproportionately over-represented in the training data (Koçak et al., 2024). Notably, there is no universally robust training set. Although the training distribution is representative of the actual test distribution, the trained model may still perform poorly on certain subgroups. In

case of a classification problem, this can be caused by the majority and minority population having different classification boundaries (Słowik & Bottou, 2021).

### 2.1.3 Annotation Bias

Label bias, or annotation bias, arises from significant variability among annotators when classifying or delineating regions of interest in diseased areas on data like medical images (Banerjee et al., 2023). As illustrated in Figure 2(b), a sensitive attribute $S$ can influence the annotation process so that the observed label $\bar{Y}$ differs from the ground truth label $Y$. Annotation shift, where certain subgroups may be systematically mislabelled more frequently than others, *e.g.* underdiagnosis of conditions in minority populations or tending to assign positive labels for ambiguous findings in elderly patients as in breast cancer screening (Autier et al., 2017), can introduce subgroup-specific inconsistencies potentially leading to annotation bias. In case of annotation shift, the model is unlikely to perform well across different subgroups, as the relationship between disease labels and imaging features becomes inconsistent (Bernhardt et al., 2022).

## 2.2 Algorithmic Bias

Apart from training data, model design plays a crucial role with respect to bias amplification. Algorithms are not impartial, and certain design choices lead to fairer prediction outcomes (Hooker, 2021). In general, ML systems that focus on minimising average error have been found to perform inconsistently across significant subsets of the data. Optimising for the loss averaged over the entire population can easily result in models that perform poorly on specific subpopulations (Słowik & Bottou, 2021). Model design choices aimed at maximising test-set accuracy often fail to preserve other important properties, such as robustness and fairness. One key reason these choices amplify algorithmic bias is that fairness often coincides with how the model treats underrepresented protected features. The algorithmic bias a model acquires can be linked to the disproportionate over- or underrepresentation of a protected attribute within a specific category. Identifying which model design choices disproportionately increase error rates for protected, underrepresented features is a critical first step in reducing algorithmic harm (Hooker, 2021). Hooker et al. (2020; 2019) ~~found~~ show that compression techniques like quantisation and pruning disproportionately impact low-frequency attributes such as age and gender in order to maintain performance on the most frequent features. Jiang et al. (2020) ~~demonstrated~~ demonstrate that difficult and underrepresented features are learned later in the training process and that the learning rate influences what the model learns. Therefore, early stopping and similar hyperparameter choices disproportionately affect a subset of the data distribution. Some model design choices are better than others regarding fairness considerations. For instance, with the widespread use of compression and differential privacy techniques in sensitive areas like healthcare diagnostics, understanding the error distribution is crucial for assessing potential harm. In such settings, pruning or gradient clipping may be unacceptable due to their impact on human well-being (Hooker, 2021). A special case of algorithmic bias is stemming from shortcut learning, which is described in more detail in the subsequent ~~section~~ Section 2.2.1.

### 2.2.1 Shortcut Learning

Shortcut learning or confounding bias emerges when ML models rely on confounding variables to derive predictions (Boland et al., 2024).

Spurious features can lead to shortcut learning, where ML models depend on superficial or irrelevant features. These features are simple for the model to learn but do not generalise beyond the training data, where the connection between the label $Y$ and the spurious feature $S$ no longer holds, leading to a drop in performance after deployment, see Figure 3. In these cases models may depend on spurious features even when they are less predictive than clinically relevant features (Boland et al., 2024). For instance, ML models relying on portable Intensive Care Unit (ICU) radiographic markers as proxies for the condition, rather than identifying the true underlying pathology for pneumonia prediction (Zech et al., 2018) or pneumothorax detection models relying on inserted chest tubes for prediction (Rueckel et al., 2020).

Conversely, ML models might use sensitive attributes $S$, such as age, sex, or race, to enhance performance, which could be justifiable when these attributes are correlated with disease risk in the target population.

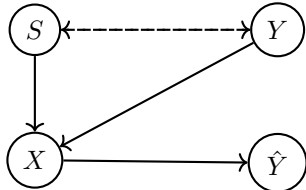

Figure 3: Graphical model for shortcut learning: input $X$ depends on both sensitive attribute $S$ and label $Y$, while prediction $\hat{Y}$ depends on $X$. The dashed bidirectional edge between $S$ and $Y$ indicates potential correlation in the dataset (*e.g.* age correlated with disease prevalence), which may lead the model to rely on $S$ as a shortcut for predicting $Y$, rather than learning clinically meaningful features.

For instance, melanoma is more common in lighter skin tones, breast cancer is more prevalent in women, and androgenetic alopecia is more common for men. In these situations, disregarding or removing attribute information ~~could~~ can reduce clinical performance (Brown et al., 2023).

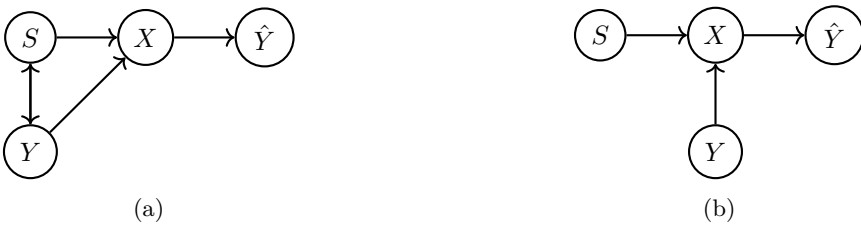

    (a)                                   (b)

Figure 4: Graphical models showing relationships between sensitive attributes ($S$), data ($X$), labels ($Y$), and predictions ($\hat{Y}$)(Dehdashtian et al., 2024). Scenario (a): $S$ and $Y$ are dependent (*e.g.* age affects both label and data). Scenario (b): $S$ and $Y$ are independent (correlation is spurious).

Figure 4 represents the two distinct scenarios of biases relating to sensitive attributes. In Figure 4(a), $Y$ and $S$ are inherently dependent, *e.g.* an attribute like age can influence both the risk of developing a condition and the appearance of the image. A model might learn to predict the presence of a condition based on the attribute. However, this scenario will introduce a trade-off between performance and fairness. In Figure 4(b), $Y$ and $S$ are independent. According to Dehdashtian et al. (2024), any observed correlation can be considered as a spurious correlation, where these correlations are not considerably beneficial for the performance of the model. In scenario (b), the performance of a bias-free model regarding $Y$ being independent of $S$ is expected.

From an ML perspective, biases can be understood as arising from dependencies between data attributes (confounding variables). The data $X$ depends on the target attribute $Y$ and the sensitive attribute $S$. Bias mitigation aims to guarantee that the prediction $\hat{Y}$ is statistically independent of $S$. To grasp the effects of biases, it is useful to distinguish between demographic and non-demographic biases. Demographic biases arise when models perform differently across various demographic groups, which can be defined by attributes like gender, race, or age. Ideally, a bias-free model should exhibit consistent performance regardless of these attributes (Dehdashtian et al., 2024). In case of a correlation between $S$ and $Y$, a performance-fairness trade-off is introduced. Non-demographic biases, *e.g.* measurement artefacts (motion artefacts in MRI), are unrelated to demographic factors (Spisak, 2022). These biases involve spurious correlations that computer vision systems can learn to solve a task. Although these biases are not tied to specific attributes, such attributes may often be identified in various tasks (Dehdashtian et al., 2024).

### 2.3  Metrics for Bias Assessment

Model fairness is commonly assessed through two principal frameworks: *individual fairness*, which ~~emphasizes~~ emphasises consistency at the instance level (Dwork et al., 2012), and *group fairness*, which evaluates outcomes across demographic or social subgroups (Barocas et al., 2023). In the context of group fairness, model behaviour is typically formalised using three core statistical criteria, each of which captures a distinct condi-

tional independence relationship among the predicted outcome $\hat{Y}$, the true label $Y$, and a sensitive attribute $S$:

- *Independence* ($\hat{Y} \perp\!\!\!\perp S$) requires that the prediction be uncorrelated with group membership. This criterion reflects equality of outcomes across groups, regardless of the ground truth label.

- *Separation* ($\hat{Y} \perp\!\!\!\perp S \mid Y$) requires that predictions be conditionally independent of sensitive attributes given the true label, ensuring equal error rates across groups.

- *Sufficiency* ($Y \perp\!\!\!\perp S \mid \hat{Y}$) requires that, for any given predicted label, the likelihood of the true label given a predicted label is consistent across groups, reflecting equal predictive value.

Each criterion corresponds to a class of fairness metrics that capture specific types of statistical dependencies (Marcinkevics et al., 2022; Castelnovo et al., 2022). For instance, the *Statistical Parity Difference* (SPD) (Savani et al., 2020), defined in Equation 1, measures the difference in the rates of positive predictions across groups, irrespective of ground truth. This metric operationalises the independence criterion, as it requires prediction rates to be equal across groups regardless of the true label.

$$SPD = P(\hat{Y} = 1 \mid S = 0) - P(\hat{Y} = 1 \mid S = 1) \tag{1}$$

The *Equality of Opportunity Difference* (EOD) (Hardt et al., 2016; Savani et al., 2020), formalised in Equation 2, quantifies the discrepancy between the True Positive Rates (TPRs) of a classifier across groups defined by the sensitive attribute $S$ (Marcinkevics et al., 2022). A significant TPR disparity indicates that individuals with a disease within a protected subgroup are not receiving correct diagnoses at the same rate as the general population, even if the algorithm has a high overall accuracy (Boland et al., 2024). EOD ~~formalizes~~ formalises a relaxed form of separation by focusing only on the positive class ($Y = 1$), thereby enforcing equal true positive rates across groups.

$$EOD = P(\hat{Y} = 1 \mid Y = 1, S = 0) - P(\hat{Y} = 1 \mid Y = 1, S = 1) \tag{2}$$

The *Predictive Parity Difference* (PPD), defined in Equation 3, measures the deviation from the predictive parity condition, which requires equal Positive Predictive Value (PPV) across groups (Chouldechova, 2017). This condition is a relaxed instance of the broader *sufficiency* criterion, which also encompasses equality of Negative Predictive Value (NPV) across groups.

$$PPD = P(Y = 1 \mid \hat{Y} = 1, S = 0) - P(Y = 1 \mid \hat{Y} = 1, S = 1) \tag{3}$$

Disparities in PPD indicate that the model's positive predictions are not equally trustworthy across groups, which can be critical when decisions depend directly on predicted outcomes. Notably, in clinical settings where false negatives carry significant consequences, *e.g.* ruling out cancer, ensuring parity in NPV may be just as important as PPV. For a more comprehensive list of metrics, we refer the reader to related surveys (Verma & Rubin, 2018; Castelnovo et al., 2022; Rabonato & Berton, 2024).

## 2.4 Challenges in Bias Assessment

The three foundational fairness criteria (*i.e.* independence, separation, and sufficiency) are notably mutually incompatible (Castelnovo et al., 2022; Barocas et al., 2023; Gao et al., 2024). As illustrated in Example 2.1, satisfying one often necessitates violating another, particularly when the predicted outcome $\hat{Y}$, true label $Y$, and sensitive attribute $S$ exhibit complex statistical dependencies. As a result, fairness assessment is inherently context-dependent and must consider which kinds of errors are more harmful in the specific application domain.

Notably, disease prevalence often correlates with demographic variables such as age, sex, or ethnicity, making rigid notions of group-level equality problematic in clinical applications (Gao et al., 2024). In such cases, enforcing fairness criteria like independence ($\hat{Y} \perp\!\!\!\perp S$), which requires prediction rates to be equal across groups, can lead to clinically inappropriate decisions, as it disregards meaningful and medically justified

differences in disease prevalence between groups. As such, defining fairness purely in terms of "equality" may be misleading (Liu et al., 2023a). Compounding this, many medical imaging datasets lack adequate socio-demographic representation, limiting the ability to assess bias comprehensively, particularly in intersectional subgroups (Stanley et al., 2023). Moreover, subgroup choice itself plays a critical role in the effectiveness of fairness interventions (Alloula et al., 2025). In some cases, mitigation strategies based on observed disparities within a particular set of subgroups can paradoxically worsen outcomes. Lastly, fairness metrics can be sensitive to distribution shifts, making their behaviour unstable over time and further complicating their interpretation in real-world, dynamic settings (Mienye et al., 2024). Together, these factors highlight the need for fairness frameworks in healthcare that are context-aware, data-driven, and aligned with both clinical utility and ethical considerations.

> **Example 2.1: Fairness Trade-Off**
>
> In breast cancer screening, models are often evaluated on their ability to detect cancer early. Suppose the model achieves equal true positive rates (TPRs) across racial groups, **satisfying Equality of Opportunity (EOD)**, a separation-based metric. This ensures that all women who have cancer are equally likely to be correctly identified, regardless of race.
>
> However, achieving this may require lowering the decision threshold for groups with historically lower TPRs, which can increase false positive rates. As a result, the Positive Predictive Value (PPV), *i.e.* the likelihood that a positive prediction truly reflects cancer, may differ across groups. This **violates the Predictive Parity Difference (PPD)**, a sufficiency-based criterion.
>
> In this case, prioritizing EOD reflects a clinical choice to reduce underdiagnosis, accepting a potential increase in overdiagnosis. This trade-off is often justified in high-risk screening contexts where missing a diagnosis is more harmful than additional follow-up testing.

## 2.5 Relevance of Bias Assessment in Clinical Practice

Bias in machine learning models has direct consequences beyond statistical fairness, particularly in sensitive domains like healthcare. Indeed, when models perform unequally across demographic groups, they can lead to unequal access to diagnosis or treatment (Obermeyer et al., 2019; Cross et al., 2024). An extensive review on bias types and examples in cardiovascular imaging (Vrudhula et al., 2024) can serve as a useful reference for understanding how such biases manifest in practice and the mechanisms through which they propagate into clinical decision-making.

Consequently, bias affects clinical workflows and outcomes, thereby reinforcing existing health inequalities. Recognising these implications highlights the importance of bias evaluation and mitigation as essential steps toward safe, fair, and robust deployment of ML systems.

# 3 Key Concepts of Data Drift

In spite of an early effort to standardize the terminology regarding dataset shift (Moreno-Torres et al., 2012), there are still multiple terms used for the same concept across literatures. These terms include domain drift, distributional shift, and dataset shift, among others (Sahiner et al., 2023). Because these phenomena and their detection have been extensively studied in general ML settings, we purposefully include non-medical work in this section to provide a principled foundation for the medical imaging applications discussed in Sect. 5.3.

In this section, we first ~~formalize~~ formalise a simplified mathematical definition of data drift. The subsequent subsections outline the motivation behind addressing the problem of data drift and the reason for the occurrence of data drift. Next, we provide insights into the different types of data drift.

The implicit assumption underlying all supervised ML techniques is that the training dataset distribution $P_{TRAIN}(X, Y)$ is the same as the distribution of the data processed by the model post-deployment

$P_{DEPLOYED}(X, Y)$. In this context, we define $X \in R^n$ as the vector representation of a data item's covariates and $Y$ its corresponding target variable (Dreiseitl, 2022). Dataset shift occurs when this assumption no longer holds, that is, there occurs a discrepancy in the joint distribution of inputs and outputs (target variables) between the training and deployment stage (Quiñonero-Candela et al., 2022) or when data changes slowly over time because of systematic errors or random population shifts. Specifically, that is the case when $P_{TRAIN}(X, Y) \neq P_{DEPLOYED}(X, Y)$.

Generally, dataset shift can be assumed whenever there are disparities between the distributions of the training and unseen data (Moreno-Torres et al., 2012). ~~(Moreno-Torres et al., 2012)~~ Moreno-Torres et al. (2012) specifies the case of sample selection bias causing dataset shift, where the aforementioned distribution discrepancies arise because the training examples were obtained through a biased method that over-represents more easily sampled population subgroups.

Regarding the term data drift, distinct definitions can be found in the literatures. ~~(Kore et al., 2024) and (Duckworth et al., 2021)~~ Kore et al. (2024) and Duckworth et al. (2021) define data drift as a "systematic shift in the underlying distribution of input features", where $P_t(X) \neq P_{t+t'}(X)$ for probability distribution $P$ defined at time $t$. In contrast, ~~(Webb et al., 2016)~~ Webb et al. (2016) summarise the term drift such that any of the elements of a joint distribution $P(X, Y)$ might be subject to change over time. Since the joint distribution can be factorized as $P(X, Y) = P(X|Y)P(Y) = P(Y|X)P(X)$, drift occurs if any of $P(X|Y)$, $P(Y|X)$, $P(Y)$ or $P(X)$ change over time. ~~(Quiñonero-Candela et al., 2022) specifies~~ Quiñonero-Candela et al. (2022) specify data drift as the circumstance in which the whole joint input-output distribution is non-stationary.

Following the definitions of data drift by ~~(Webb et al., 2016) and (Quiñonero-Candela et al., 2022)~~ Webb et al. (2016) and Quiñonero-Candela et al. (2022), we will define data drift as any occurrence of change in the joint distribution over time $P_t(X, Y) \neq P_{t+t'}(X, Y)$.

It is worth mentioning here that data drift is conceptually different from the traditional task of out-of-distribution detection and anomaly detection ~~(Soin et al., 2022), as in (Baugh et al., 2023a)~~ (Soin et al., 2022; Baugh et al., 2023a). Data drift can manifest as a gradual shift in any of the elements of the joint distribution, whereas individual outliers or anomalies might appear without a data drift. In case of drift detection, the aim is to intervene at model level (*e.g.* retrain, removal from production). On the contrary, if out-of-distribution input is identified, *e.g.* by measuring the difference between the image and its reconstruction (Müller & Kainz, 2022), the assumption is that the model still performs effectively, but was not accurate for that particular input data (Soin et al., 2022).

The most significant reasons for data drift are non-stationary environments (Moreno-Torres et al., 2012). Even if the distribution of the training data matches the data post deployment, it may still be subject to drift over time (Sahiner et al., 2023). In case of non-stationary environments, the training environment differs from the deployment environment due to temporal changes in distribution (Moreno-Torres et al., 2012) thereby violating the stationary assumption underlying ML models (Cieslak & Chawla, 2009).

### 3.1 Types of Data Drift

Following the terminology by ~~(Moreno-Torres et al., 2012)~~ Moreno-Torres et al. (2012), three different kinds of shift can appear, namely Covariate Shift, Prior Probability Shift and Concept Shift. These shifts do not necessarily have to happen independently from each other, but can happen simultaneously.

When examining dataset shift, the relation between covariates and class labels is highly relevant (Moreno-Torres et al., 2012). Specifically, following our definition of data drift as change in the joint distribution $P_t(X, Y) \neq P_{t+t'}(X, Y)$, and effectively, a change in any of $P(X|Y)$, $P(Y|X)$, $P(Y)$ or $P(X)$, causal and anticausal settings need to be distinguished ~~(Castro et al., 2020) (Schölkopf et al., 2012)~~ (Castro et al., 2020; Schölkopf et al., 2012) in context of the different types of data drifts. Understanding of these two relationships is also key to ~~the~~ understanding and distinguishing the various types of data drifts explained in the following subsections.

- Causal: Estimate $P(Y|X)$ in $X \rightarrow Y$ problems, where the class label $Y$ is causally determined by the features $X$ (*e.g.* prediction of labels $Y$ from medical images $X$) (Moreno-Torres et al., 2012; Castro et al., 2020).
- Anticausal: This is the opposite direction where the goal is to determine $P(X|Y)$ in $Y \rightarrow X$ problems, where class label $Y$ causally determines features $X$ (Moreno-Torres et al., 2012; Castro et al., 2020). ~~(Schölkopf et al., 2012)~~ Schölkopf et al. (2012) provide examples of class label prediction task from handwritten digits images. The causal structure here is: the person's intention to write a digit (say, 7) causes a motor pattern resulting in the image. Effectively, the class label $Y$ causes $X$, the image or image features.

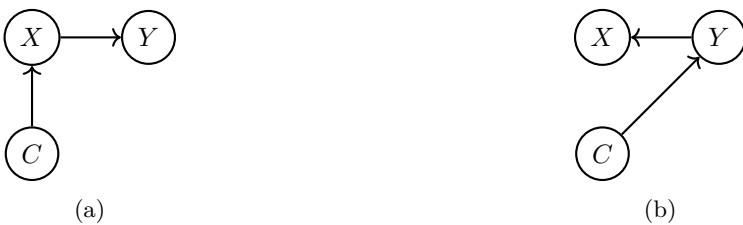

(a)                                                                                                              (b)

Figure 5: Graphical causal models illustrating causal relationships between covariates $(C)$, input $(X)$, labels $(Y)$ for (a) Covariate shift: A confounding covariate influences the input, which in turn influences the output, indicating a causal mechanism and (b) Prior Probability Shift: A confounding covariate influences the labels consequently influencing the input, thereby indicating an anticausal mechanism.

In the following subsections, we will elaborate and formalize the three common different kinds of data drifts and their corresponding causal relations.

### 3.1.1 Covariate Shift

Covariate shift appears only in $X \rightarrow Y$ problems, when $P_{TRAIN}(Y|X) = P_{DEPLOYED}(Y|X)$, but $P_{TRAIN}(X) \neq P_{DEPLOYED}(X)$ (Moreno-Torres et al., 2012). Only the covariate distribution is subject to change between the training and the distribution ~~post deployment~~post-deployment, and the model $P(Y|X)$ remains unaffected. However, $P_{TRAIN}(X) \neq P_{DEPLOYED}(X)$ should not be mistaken as implying that the rule for the prediction of $Y$ from $X$ does not need to be adapted to the new covariate distribution $P_{DEPLOYED}(X)$. This is reasoned by the fact that predictions based on finite data may favour simple functions that perform well in regions where $P_{TRAIN}(X)$ is high, but not where $P_{DEPLOYED}(X)$ is high (Schölkopf et al., 2012). In the context of causality, by definition, we can remark that covariate shift is associated with causal mechanisms or $X \rightarrow Y$ scenarios where the features cause the label, see Figure 5a.

In case of acquisition shift or domain shifts, the measurement system or method of description is subject to change (Quiñonero-Candela et al., 2022). In medical imaging, domain shift is responsible for potentially harmful disparities between development and deployment conditions of medical image analysis techniques. In this context, acquisition shift relates to variations in the likelihood $P_D(X|Z)$ of an image $X$ obtained from a particular domain $D$, *e.g.* its appearance, given the unobserved, latent reality $Z$ of a patient's true anatomy (Hognon et al., 2024). Domain or acquisition shift can be understood as a change in the mapping function $X = f(Z)$, where the target variable $Y$ is dependent on the latent, never directly observable variable $Z$ (Quiñonero-Candela et al., 2022). Since a change in the mapping function has an effect on the covariate $X$, these types of shifts can be seen as subcategories of covariate shift.

In the context of medical imaging, a common cause for covariate shift are different image acquisition devices, *e.g.* multiple manufacturers of scanners or acquisition protocols in clinical use, leading to varying qualities of images. Patient populations subject to change over time also constitute covariate shift (Sahiner et al., 2023), *e.g.* when a new disease appears during a pandemic.

### 3.1.2  Prior Probability Shift

Prior probability shift occurs only in $Y \to X$ problems or in anticausal settings, when $P_{TRAIN}(Y) \neq P_{DEPLOYED}(Y)$, but the relationship $P_{TRAIN}(X|Y) = P_{DEPLOYED}(X|Y)$ remains (Moreno-Torres et al., 2012), see Figure 5b. The terms label shift and prevalence shift can be used interchangeably for prior probability shift (Castro et al., 2020; Garg et al., 2020).

In case of label shift in diagnostic problems where diseases cause symptoms, the optimal predictor might be subject to change, *e.g.* during a pandemic, the probability of a patient having a disease given their symptoms can increase (Garg et al., 2020). Suppose $Y$ is the target that denotes the probability of a disease with $P(Y = 1) = 0.25$ and $P(Y = 0) = 0.75$. In the event of a pandemic, the disease probability may increase, giving rise to a new probability distribution $P(Y = 1) = 0.5$ and $P(Y = 0) = 0.5$. Although the conditional probability of the disease in the case of a symptom $X$ remains unchanged, *i.e.* $P_{TRAIN}(X|Y) = P_{DEPLOYED}(X|Y)$, the joint distribution $P(X, Y)$ changes due to change in $P(Y)$, causing by definition a data drift induced by prior probability shift.

### 3.1.3  Concept Shift

Concept shift (Eq. 4) occurs when there is a change in the relationship between $X$ and $Y$. Following ~~(Moreno-Torres et al., 2012)~~Moreno-Torres et al. (2012), it can be defined as:

$$P_{TRAIN}(Y|X) \neq P_{DEPLOYED}(Y|X), \text{ but } P_{TRAIN}(X) = P_{DEPLOYED}(X) \text{ in } (X \to Y) \text{ settings}$$
$$P_{TRAIN}(X|Y) \neq P_{DEPLOYED}(X|Y), \text{ but } P_{TRAIN}(Y) = P_{DEPLOYED}(Y) \text{ in } (Y \to X) \text{ settings} \quad (4)$$

Thus, concept shift can occur in both causal and anticausal settings whenever the conditional relation between the cause and effect or the input and target variables changes, see Figure 6. An example of concept shift in causal setting can be seen in the following: After 2020 certain patterns of patchy ground-glass opacity in chest X-rays might not be labelled as bacterial pneumonia anymore, but as COVID-19 pneumonia (Sahiner et al., 2023).

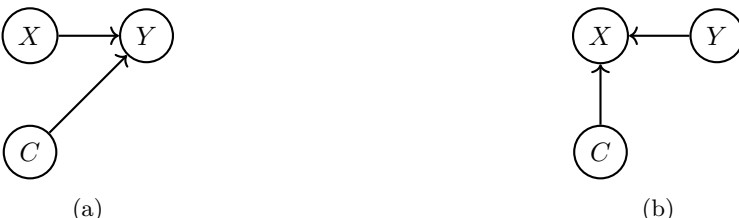

Figure 6: Graphical causal models illustrating causal relationships between confounding covariates ($C$), input ($X$), labels ($Y$) for two distinct cases of concept shift, namely, (a) Concept shift in causal settings: The confounding covariate influences the output, thereby influencing the initial relation between inputs and labels (b) Concept shift in anticausal settings: The confounding covariate influences the input thereby changing the initial anticausal relation.

### 3.2  Data Drift Assessment and Inherent Challenges

In this subsection, we present several common methods to assess data drift, their current limitations, and the inherent challenges that arise when using these methods.

Data drift assessment can be performed by comparing several commonly used machine learning assessment metrics between the original data domain and the drifted data domain. Performance metrics such as accuracy and segmentation DICE scores are used to determine the performance drop induced by dataset shifts. Uncertainty metrics can be used to determine and identify data drifts, ideally with higher scores signifying higher uncertainties that arise from data drifts:

- *Confidence:* Quantifies the model's uncertainty through the predicted probability of the top class.

- *Entropy:* The sum of the negative log of the prediction probability distribution across all classes.

- *Calibration:* The absolute difference between the model's confidence and the prediction empirical accuracies across all classes.

One can measure the domain gap between the original train and test set with several common distributional distance metrics:

- *Fréchet Inception Distance (FID):* Distributional distance between two distinct distributions given their mean and variance.

$$D_F = \|\mu - \mu_{TEST}\|_2^2 + \text{Tr}(\Sigma + \Sigma_{TEST} - 2(\Sigma\Sigma_{TEST})^{\frac{1}{2}}), \tag{5}$$

  where $\mu$ represents the mean vector and $\Sigma$ denotes the covariance matrix of the respective data sets. Tr denotes the trace operator.

- *Maximum Mean Discrepancy (MMD):* Divergence between kernel mean embeddings of the datasets.

$$\text{MMD}^2 = \mathbb{E}[k(\mathbf{x}_{TRAIN}, \mathbf{x}'_{TRAIN})] + \mathbb{E}[k(\mathbf{x}_{TEST}, \mathbf{x}'_{TEST})] - 2\mathbb{E}[k(\mathbf{x}_{TRAIN}, \mathbf{x}_{TEST})], \tag{6}$$

  where $\mathbf{x}_{TRAIN}, \mathbf{x}_{TEST}$ are samples from training and test distributions, and $k$ is a kernel, for example, a Gaussian kernel.

However, there are inherent challenges that hinder the direct applicability of the aforementioned metrics. Performance based methods necessitate ground truth, which might be difficult to obtain in a timely manner or might not be available in general (Kupinski et al., 2002). Uncertainty metrics such as entropy and confidence may also not be reliable as demonstrated by ~~(Ovadia et al., 2019)~~Ovadia et al. (2019). Direct embedding of data distributions can also be infeasible for real-time drift detections due to the large compute requirements for data of higher dimensions such as higher resolution images. Compared to bias assessment methods, drift assessment methods found in the literature employ a wider variety of other methods and metrics to remedy the potential limitations. We elaborate these methods further in section 5.

### 3.3   Relevance of Data Drift Detection in Clinical Practice

Data drift can lead to malfunction or performance deterioration of ML models (Sahiner et al., 2023). In the case of classifiers, the negative effect data drift could potentially have on a classifier's performance is caused by a change in the optimal decision boundary (Moreno-Torres et al., 2012). Further, the shifted feature distribution may primarily fall into a region where the model performs poorly (Duckworth et al., 2021).

Therefore, if data drift is detected, a model performance re-evaluation given the current data is necessary, especially in high-stakes scenarios like healthcare (Kore et al., 2024). In case of performance deterioration, appropriate action needs to be taken, such as retraining the model (Duckworth et al., 2021). In the context of healthcare, early detection ensures that reliable medical care can be provided for patients (Kore et al., 2024). Monitoring the model's performance post-deployment at the output level by comparing the model's output with the ground truth labels is attractive, but is often infeasible due to a lack of ground truth information in ML stratified workflows (Dreiseitl, 2022). Automatic data drift detection can alert operators and allow them to take appropriate actions in terms of model retraining, replacement etc. (Kore et al., 2024). In cases where a drift was detected, but the model's performance did not deteriorate, these insights can be useful in understanding the generalizability of the model to new populations (Kore et al., 2024). Further, it is of high interest to determine the type of changes that occurred between the training and the post-deployment situation (Quiñonero-Candela et al., 2022).

### 3.4 Boundary between Bias and Drift

The preceding sections treated bias (Sect. 2) and data drift (Sect. 3) as conceptually distinct phenomena arising at different stages of the model lifecycle. However, bias and drift can be expressed within a single decomposition of the joint distribution. At deployment time $t$, the discrepancy between training and deployment distributions can be defined as:

$$\Delta_t = P_{DEPLOYED,t}(X,Y) - P_{TRAIN}(X,Y) \tag{7}$$

During training, dataset bias can be introduced through data acquisition, selection, annotation, or the model being trained. During deployment, the model is exposed to further changes, such as shifts in patient populations, acquisition protocols, or clinical routines. In simple terms, dataset bias characterises how the model *starts* at $t = 0$, while data drift describes how the environment *moves away* from that starting point over time. Together, they form complementary contributions to the overall shift between training and deployment conditions.

Additionally, we would like to bring up the intersections between bias and data drift from two examples previously mentioned within this work. First example is the acquisition bias arising from scanner hardware and imaging protocols (Guan & Liu, 2021). Assuming no other time dependent data drift, we illustrate the causal relationship of the acquisition bias in Figure 7a. The causal relationship is similar to covariate shift defined in 5a. Another example is the selection shift (Moreno-Torres et al., 2012) introduced in 3. The anti-causal relationship is illustrated in Figure 7b. Note how the relationship is similar to prior probability shift, even though prior probability shift takes into account the continuous deployment time, while sample selection bias is modelled as the bias that already occurs at $t = 0$.

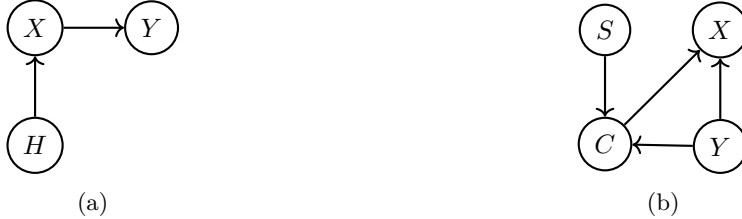

| (a) | (b) |
|---|---|

Figure 7: Graphical causal models illustrating causal relationships between hardware ($H$), confounding sensitive attributes ($S$), selecting mechanism ($C$), input ($X$), labels ($Y$) for (a) Acquisition bias: Imaging hardware selection influences the input, which in turn influences the output, indicating a causal mechanism and for (b) Sample selection bias: Sensitive attributes and labels influencing the selection mechanism, which together with the labels directly influence the input, indicating an anticausal relationship.

## 4 Methods for Bias Encoding

This section reviews various approaches that uncover or analyse biases in medical imaging models. The aim is to identify inherent biases that compromise fairness and generalisation for disparate subgroups, and to detect biased models (Seyyed-Kalantari et al., 2020; 2021). The section is ~~organized~~ organised into: assessing the presence and influence of predefined confounders in machine learning models (Sect. 4.1), assessing unknown confounders via causal inference (Sect. 4.2), and creating datasets for the structured evaluation and comparison of confounder assessment (Sect. 4.3).

### 4.1 Bias Encoding Assessment

The following methods provide insight into whether bias has been introduced into medical imaging models through confounder variables. We first focus on demographic confounders, such as age, sex, or race. Since clinics can have patients of greatly varying demographics, such confounders can have an immense detrimental impact on clinical practice, especially for minority groups. We then focus on the broader concept of non-demographic confounders, which might affect model performance due to more general forms of shortcut

learning. As discussed in Sect. 2, non-demographic confounders can introduce new demographic biases or reinforce existing ones.

### 4.1.1 Demographic Confounders

**Detection of Encoded Bias**   In deep learning architectures, the final layer of a classifier bases its prediction on the features computed by the previous layers of the model. However, how can we determine which information encoded in the features actually leads to the final prediction of the model? To investigate this, we can test whether the information encoded in its features enables the model to solve an unrelated task, such as race prediction. This shows us if additional racial information unrelated to the cancer detection task is present in the features. If the accuracy is high for the race detection task, then this information has to be present in the features. This approach ~~was~~ has been used by Glocker et al. (2023a), based on the study of Gichoya et al. (2022). They found that a disease classifier backbone trained on a single race has the same race classification results as a backbone without this restriction, which implies that unrelated racial information is present in the features.

However, this does not inform us whether this information is actually *used* for cancer detection, only that it is *present*. We need to employ other methods to detect whether the encoded bias facilitates harmful shortcut learning (see Sect. 2.2.1.)

**Influence of Encoded Bias**   There currently exist two frameworks to predict the influence of encoded biases on the decision making processes in medical imaging: one analysing whether confounders influence the separability of features (Glocker et al., 2023a), the other measuring the correlation between encoding strength and fairness (Brown et al., 2023).

The framework introduced by Glocker et al. (2023a) applies a dimensionality reduction, *e.g.* PCA, to the features of the penultimate layer of a disease classifier. The rationale ~~being~~ is that the penultimate layer contains the learned features of the model, while the final layer decides which of the encoded features to use for the task. Given a model trained for disease detection, the strongest separation of samples with and without disease can be found in the first few PCA modes. However, if these also separate other attributes well, then the decision of the model is likely influenced by them. This framework ~~was~~ has been applied in two follow-up studies: Piçarra & Glocker (2023) applied this framework to detect biases in generated features in age prediction models based on brain imaging data. They found that some features, which are valuable for age prediction, also contribute to distinguishing between racial and biological sex subgroups. Additionally, Glocker et al. (2023b) used their developed framework (Glocker et al., 2023a) to analyse a recently published chest radiography foundation model (Sellergren et al., 2022) for the presence of biases. An example of how the framework can be applied for this task is illustrated in Example 4.1. Statistically significant differences were observed in ten out of twelve pairwise comparisons across biological sex and race in the foundation model studied, demonstrating racial and sex-related bias. Consequently, this framework allows practitioners, who might have limited insights into foundation model pretraining, to thoroughly assess bias encoding in foundation models applied to downstream tasks (Glocker et al., 2023b).

---

**Example 4.1: Feature Analysis**

**Principal Component Analysis (PCA)** rotates the feature space into orthogonal directions (principal components) ordered by their variance. Projecting data onto the first few components preserves the strongest signals while reducing dimensionality.

**Kolmogorov-Smirnov (KS) tests** are non-parametric tests that compare the empirical cumulative distributions of two samples and return a p-value indicating whether they stem from the same underlying distribution.

**Approach:** First, we train the network on radiographs (annotated with disease, sex, race). For every image $i$, we store the penultimate layer vector $\mathbf{h}_i$.

Then, we compute the PCA of our features and compute its first four principal components, yielding coordinates $\mathbf{z}_{i1\ldots4}$.

Finally, for each protected attribute and each of the first two PCs, we perform a two-sample Kolmogorov–Smirnov (KS) test.

**Example results** (significant results in **bold**):

| Comparison | $p$ (PC1) | $p$ (PC2) |
|---|---|---|
| Disease vs. healthy | $1.3 \times 10^{-12}$ | 0.42 |
| Male vs. female | **0.007** | 0.52 |
| White vs. Black | 0.23 | 0.18 |

**Conclusion:** PC1 cleanly separates disease *and* sex, suggesting the model may exploit sex-specific information (shortcut learning). PC2 appears unrelated to either protected attribute.

---

The second framework by Brown et al. (2023) assesses the encoding of sensitive attributes, analyses fairness metrics and introduces shortcut testing (ShortT). This provides insights into the correlation between the encoding of the sensitive attribute and fairness metric to investigate how shortcut learning might affect model fairness and performance. The underlying assumption is that the influence of sensitive attributes on the model consists of both biological, potentially causal effects that could improve model performance (graph (a) in Figure 4), and shortcut learning that could be harmful (graph (b) in Figure 4). Even though both scenarios introduce bias, Brown et al. (2023) only refer to shortcut learning when a sensitive attribute is used as a confounder, which does not significantly enhance performance, but impacts fairness. To assess the encoding of sensitive attributes such as age in their analysis, they trained a model for disease prediction, froze all weights in the model backbone and then trained a predictor for age using a mean squared error (MSE) regression loss. The performance of the transfer model was measured using the Mean Absolute Error (MAE), which serves as an indicator of the age-related information captured by the final layer of the feature extractor. Lower MAE values correspond to more accurate age predictions, indicating stronger age encoding. To quantify the fairness of the model's disease predictions in relation to age, Brown et al. (2023) calculated the separation metric. Separation is defined by fitting two logistic regression models to the binarised model predictions. To this end, the patients were split by the true label: $Y = 1$ (has the condition) and $Y = 0$ (does not). In each split, they fitted a logistic regression that predicts whether the model outputs a positive prediction ($\hat{Y} = 1$ based on age. In the $Y = 1$ group, this tracks how the True Positive Rate (TPR) varies with age, while in the $Y = 0$ group this tracks how the False Positive Rate (FPR) varies with age. Let $\beta_1$ be the age coefficient in the $Y = 1$ regression and $\beta_0$ the coefficient in the $Y = 0$ regression. Separation is defined as $Sep = \frac{1}{2}(|\beta_1| + |\beta_0|)$. Values near 0 mean TPR and FPR do not systematically change with age, while larger values mean they do. To assess the degree of age encoding on model fairness, they used a multitask learning model by adding an age prediction head to the base model (disease prediction model) and scaled the gradient updates from the age prediction head. For each value of gradient scaling, they computed the model disease prediction performance, MAE of the age prediction and separation fairness metric. Shortcut learning was indicated by a significant correlation between age encoding and separation fairness metrics (computed via Spearman correlation coefficient). The proposed framework represents a feasible approach for analysing the impact of shortcut learning since the analysis only involves the addition of a demographic prediction head to the base model. Practitioners need to carefully consider which fairness metric to select (besides separation, *e.g.* demographic parity (independence) could be selected), and whether the gradient

intervention consistently modifies the encoding of the protected attribute before evaluating its relationship with model fairness (Brown et al., 2023).

### 4.1.2 Non-Demographic Confounders

The methods proposed below assess the model encoding of non-demographic biases. Such confounders may include any feature in the image that correlates well with a class (see Sect. 2.2.1 for examples). We first give an overview of an important family of non-demographic confounders in biomedical imaging: characteristics introduced during the image acquisition, as introduced by specific CT scanners, for example. These works demonstrate that even low-level acquisition cues are perfectly predictable from pixels and can therefore be subject to erroneous shortcut learning. Then we move from discussing specific acquisition-related factors to studies that either create synthetic shortcuts or mask out possible shortcut regions. These methods can be used to detect the presence of any kind of pixel-level confounder.

**Shortcuts via Acquisition Parameters** Lotter (2024) investigated how acquisition parameters such as view position, field of view, or window size affect race classifiers based on radiographs. They found that changing these parameters strongly ~~influenced~~ influences model performance, *e.g.* using only lateral view radiographs increases the performance for Asian patients, but decreases the performance for detecting Black patients. They used this insight to mitigate the disease detection performance gap in the baseline dataset between White patients and minorities ~~,~~ by introducing a unique classification threshold for each view position, respectively.

In a related study, Badgeley et al. (2019) showed a strong example of how radiograph classifiers can largely base their decisions on acquisition parameters rather than pathology. They found that a classifier trained to detect hip fractures using only image data was able to confidently predict both patient-related and acquisition-related metadata. Although the image-only model achieved strong hip fracture classification on the standard data, the performance dropped to pure chance when the fracture vs non-fracture metadata characteristics distributions were equalized. A complementary experiment showed that a classifier only trained on characteristics performed better than the image-only model. The best result were obtained by combining both image data and the characteristics during training.

**Shortcut Modification** Boland et al. (2024) used Prediction Depth (PD) to detect shortcut learning after adding synthetic shortcuts to the training data. PD measures the example difficulty, *i.e.* the computational effort a trained network needs to make a prediction for a specific input, as the number of layers a model requires to make a final prediction. Boland et al. (2024) linked PD to shortcut learning and the simplicity bias in neural networks, ~~which was~~ originally proposed by Murali et al. (2023), who showed that shortcuts are harmful when they are simpler than the relevant features. To test their method, they trained two binary classification models, one with no shortcuts and the other one with synthesised shortcuts (*e.g.* small red squares, or curved lines added to images), that ~~were~~ are perfectly correlated with one class. Both models were evaluated on the same test set, with the shortcuts balanced equally between the two classes. To verify the hypothesis that shortcuts reduce PD, the Welch's t-test was applied to assess whether there ~~was~~ is a statistically significant difference in mean PD between the two models. One dataset in their analysis was the CheXpert medical imaging data set (Irvin et al., 2019), where they also introduced synthetic shortcuts to the data. Since the method was only tested on synthetically created shortcuts, its effectiveness on natural shortcuts, such as site-specific shortcuts in multi-site data, remains to be evaluated. Additionally, the current method has a limitation in that it always requires a dataset without shortcuts as a reference.

In a related work, Sourget et al. (2025) examined whether the model ~~relied~~ relies on clinically non-relevant parts in radiographs and eye fundus images by systematically masking out irrelevant regions of the image and evaluating disease detection performance. In particular, they trained five different model types: using the unmodified images, masking out the entire region of interest (ROI) bounding box, masking out the exact ROI, masking out everything outside the ROI bounding box, and masking out everything outside the exact ROI. They found that models trained on full images ~~performed~~ perform better when evaluated on images with the ROI masked than when evaluated on images containing only the ROI. Additionally, models trained only on images with the ROI masked still achieved high classification scores. Both results indicate that the

models ~~used~~ use features that are not clinically relevant for classification. This result is further exemplified by their experiment in which a trained radiologist was asked to classify radiographs with masked-out ROIs. Unlike the models, the radiologist struggled with this task, which further supports the assumption that the models use characteristics that are not considered relevant by professionals.

Section 4.1 has shown how to detect and quantify the presence and influence of specific, predefined attributes on the decision-making of the model. However, in many cases, the confounding variable is unknown and needs to be characterised as a latent variable. One method for assessing the influence of unknown confounders, ~~casual~~ *causal* inference, is discussed in the next session.

## 4.2 Confounding Bias Detection Through Causal Inference

Unlike specifically chosen confounders, such as age or sex, many unknown confounders might be present in the data or in the learned model weights. As demonstrated in Example 4.2, causal inference provides methods to detect such unknown confounders.

> **Example 4.2: Causal Inference**
>
> Assume we have detected a correlation between biological factors $\mathbf{X}$ (*e.g.* age) and a single measure derived from radiographs $Y_i$ (*e.g.* bone age score). How can we rule out a confounding structure $\mathbf{X} \leftarrow \mathbf{Z} \rightarrow \mathbf{Y_i}$ and determine a definite causal relationship between the two? This is the question that causal inference tries to answer. One approach is to use the minimal description length principle: we fit the competing hypotheses and calculate the probability that each assigns to the observed data after averaging over all parameter settings. The hypothesis with the lower description length, *i.e.* the one with the smaller negative log-evidence, is judged the more plausible explanation of the observed correlation.

Wachinger et al. (2019) introduced a method to distinguish between causal and confounded relationships using causal inference. Confounding, with respect to the neuroimaging dataset used in their analysis, refers to imaging site-specific information being implicitly encoded by the model. Notably, the confounders are treated as unknown, latent variables in their method, which is beneficial in complex neuroimaging studies (and also in other medical imaging datasets) where the assumption of causal sufficiency (*i.e.* knowing all confounding variables) is often violated in practice. Their method aims to assess whether $X$ is more likely to cause $Y$, or if an unobserved random variable $Z$ is the underlying cause of both. Their method leverages the algorithmic Markov condition (AMC), which asserts that if $X$ causes $Y$, the factorisation of the joint distribution $P(X, Y)$ in the true causal direction will have a lower Kolmogorov complexity than the reverse, anti-causal direction. To approximate Kolmogorov complexity, which is not directly computable, the minimum description length ($L$) is applied. The factorisation of the causal scenario $P(X, Y)_{ca} = P(Y \mid X)P(X \mid Z)P(Z)$ is represented by a linear regression model, whereas the factorisation of the confounded scenario $P(X, Y)_{co} = P(Y \mid Z)P(X \mid Z)P(Z)$ is estimated by probabilistic PCA. To compare the causal with the confounded model $\Delta(X, Y) = L_{co}(X, Y) - L_{ca}(X, Y)$ is computed. If the causal model provides a better fit to the data than the confounded model, then $\Delta(X, Y) > 0$. It has to be noted that inferring causality from observational data is difficult because the true causal effects and their magnitudes are unknown, making quantitative evaluation impossible.

While the previous methods have allowed us to gain some insight into the influence certain data attributes have on a model, we currently lack evidence of how well these methods perform under specific conditions, systematic ways to evaluate the effect of different dataset characteristics, and a level playing field on which to compare bias assessment algorithms. For this purpose, dataset creation methods have been developed.

## 4.3 Representative Datasets For Bias Assessment

The following methods focus on creating datasets to mitigate and study bias. Either by constructing datasets with controlled bias components for evaluating bias mitigation, or by removing bias directly from the dataset, so no further bias mitigation is necessary.

To objectively analyse the impact of biases on medical imaging models without the limitations of real-world datasets (*e.g.* unknown confounding factors), Stanley et al. (2023) proposed SimBA, a versatile framework for generating synthetic neuroimaging data with controlled simulation of disease, bias, and subject effects. Synthetic images were generated by applying non-linear diffeomorphic transformations to a template image $I_T$ that represents the average brain morphology. Non-linear transformations for disease and bias were spatially localised deformations to $I_T$ whereas the subject morphology was generated through a global non-linear transformation. These effects and deformations were derived from PCA-based generative models of these non-linear deformations. Each effect represented a specific degree of morphological variation within the typical range of inter-subject human brain anatomy. This method allows to generate synthetic datasets with controlled bias effects. However, it has to be noted that expert knowledge is required to define the brain regions for the localised deformation for the disease and bias effects. Furthermore, it needs to be highlighted that the introduced biases were not designed to replicate any particular real-world sociodemographic sub-population, as the imaging features that introduce bias within these groups are often complex, interrelated and/or unknown. Instead, the simulated biases create hypothetical subgroups within a dataset that exhibit specific confounding features, which may lead to shortcut learning in medical imaging ML models. However, the synthetically biased datasets provide an opportunity to test bias mitigation strategies.

Stanley et al. (2024b) extended their research on the SimBA framework (Stanley et al., 2023) by using this tool to study bias manifestations and the effectiveness of bias mitigation techniques. They ~~generated~~ generate counterfactual neuroimaging datasets with three bias scenarios (*i.e.* No Bias, Near Bias, Far Bias with Near and Far indicating the proximity to the voxels representing disease). By incorporating unequal proportions of biased brain images for the disease class and non-disease class (70% vs. 30% containing bias) they incorporated the possibility of the model ~~to use bias as~~ using bias as a shortcut for predicting the disease. At the same time they ensured that subject and disease effects ~~were~~ are similar for each bias group (preventing unwanted additional ~~source~~ sources of bias), enabling a controlled evaluation of bias. Stanley et al. (2024b) then ~~used~~ use these synthetically generated images to assess the efficiency of various bias mitigation strategies (reweighing, unlearning, group models). The SimBA framework represents an interesting approach to test models for bias encoding, but the bias scenarios that can be generated by this framework are limited to scenarios in which biases are represented by localised spatial deformations, or are related to intensity-based simulated artifacts. Real-world medical imaging data can be more complex and inherently contain numerous confounding and interacting biases.

Apart from these synthetically generated biased/unbiased datasets, Glocker et al. (2023a) used strategic resampling with replacement to create balanced test sets that represent the population of interest. The goal was not replicating dataset bias in the test set (due to random splitting of the original dataset) and to allow an unbiased estimation of performance on different race subgroups. Specifically, they used resampling with replacement to create ~~race balanced~~ race-balanced test sets while controlling for age differences and disease prevalence for each racial group.

## 5 Methods for Data Drift Detection

This section provides an overview of methods developed for data drift detection in a general setting for prevalence drift (Sect. 5.1) and covariate drift (Sect. 5.2)~~.~~, independent of a specific application domain. We deliberately draw on the broader ML literature here, since the underlying statistical notions of drift and most detection algorithms are domain-agnostic and form the basis of methods adapted to medical imaging. The following section presents a multitude of approaches in more detail, ~~focussing~~ focusing on the application in the medical imaging domain (Sect. 5.3). We follow the common categorisation into classifier-based, feature-based, and metadata-based approaches.

### 5.1 Prevalence Drift Detection

Roschewitz et al. (2023) ~~introduced~~ introduce Unsupervised Prediction Alignment (UPA), ~~which was~~ originally designed to recover the desired sensitivity/specificity trade-off in the case of acquisition shift. UPA can also be applied to detect a prevalence shift. This involves using linear piecewise cumulative distribution matching to align the prediction distribution from the unseen dataset with a reference distribution of

fixed prevalence. In the case of prevalence shift detection, the mean absolute difference between the original (shifted) and aligned predictions is calculated. When there is no prevalence shift present in the data, the difference should be 0. A limitation of this method is that it only detects shifts in prevalence, reducing its clinical practicality as it would need to be used alongside other drift detection techniques.

## 5.2 Covariate Drift Detection

### 5.2.1 Dimensionality Reduction

Rabanser et al. (2019) base their analysis of shift detection between imaging source (training) and target data (artificially shifted data) on common dimensionality reduction methods combined with two-sample tests. Dimensionality reduction methods try to compress the numerical representation of extracted features , by transforming the feature vector into a smaller one without losing important information (*i.e.* preserving structure between feature vectors of different samples). This transformation can either be learned (*e.g.* Auto-Encoders, cf. Example 5.1) or constructed analytically (*e.g.* PCA, cf. Example 4.1). To compare the differences between the source and shifted dataset after reducing their dimensions, the authors used several statistical methods, especially the Maximum Mean Discrepancy (MMD), cf. Equation 6, to compare the multivariate distributions and the Kolmogorov-Smirnov (KS) test for individual dimensions separately.

Rabanser et al. (2019) ~~used~~ use the MNIST and CIFAR-10 datasets for their analysis and simulate shifts affecting both covariate and label proportions. Their compared methods include PCA, sparse random projections, trained and untrained Auto-Encoders (tAE, uAE), label classifier, and domain classifier. They find that the Softmax outputs from the trained label classifier, combined with multiple univariate KS testing yields the best performance in shift detection, followed by multivariate-testing of an untrained Auto-Encoder embedding.

It has to be highlighted that the number of target (shifted) data samples has a substantial influence on the accuracy of the shift detection. This has to be taken into consideration when applying the shift detection methods in practice. Consequently, selecting an appropriate rolling window time frame is essential to have a sufficient ~~amount~~ number of data samples to compare with the reference (training) data. Since the shift detection based on Softmax outputs combined with KS testing can be easily added on top of an already trained classifier, this method constitutes a feasible, practical solution.

> **Example 5.1: Auto-Encoder**
>
> **Auto-Encoders** are a common and simple network architecture that draws inspiration from sparse coding and works in an unsupervised manner. The architecture consists of two parts, the Encoder and the Decoder. Both a trained in tandem while during inference, only one of the two, depending on the application, is used. The key aspect is ~~a~~ an information bottleneck between Encoder and Decoder. Specifically, the Encoder transforms the input over several non-linear layers into a ~~lower dimensional~~ lower-dimensional representation (the bottleneck), while the Decoder, acting as regularisation, has to be able to reconstruct the original input using only the compressed representation. Hence, the objective function is minimizing the reconstruction error between input and output, typically measured as the MSE.
>
> **Variational Auto-Encoders** are a probabilistic extension of the classical Auto-Encoder. Specifically, their bottleneck does not represent a deterministic lower dimensional feature space, but defines independent distributions for each compressed dimension, *e.g.* mean and variance of an assumed Gaussian distribution. These latent distributions allow the generation of new samples (images) that appear close to the training sample distribution, *e.g.* by drawing from every Gaussian in the bottleneck.

### 5.2.2 Uncertainty Estimation

Investigating the metrics discussed in Sect. 3.2, Ovadia et al. (2019) use approaches of uncertainty estimation for identifying data drift in image data. Their findings suggest that entropy scores and prediction confidence

are ineffective when tested on out-of-distribution (OOD) data (based on a class excluded from the training set in the MNIST dataset). Most calibrated models exhibit low entropy and high confidence in their study, meaning they were confidently incorrect when predicting entirely OOD data. Consequently, changes in these two metrics do not provide a reliable indication of data drift leading to increased uncertainty, as measured by entropy scores and prediction confidence. Concerning the Expected Calibration Error (ECE) they found that ECE increased as the shift intensity grew. However, since ECE depends on ground truth labels, using this measure to detect drift is not practical for automated drift detection in clinical settings.

### 5.2.3  Image Segmentation

To predict segmentation performance in the absence of ground truth labels, Valindria et al. (2017) propose reverse classification accuracy (RCA). Specifically, a classifier is trained on a single image with its predicted segmentation (where the segmentation prediction of this single image is derived from the original segmentation model) serving as pseudo ground truth (GT). The method is built on the assumption that if the segmentation quality for a new image is high, then the RCA classifier trained on the predicted segmentation used as pseudo GT will perform well at least on some of the images in the reference database. Similarly, if the segmentation quality is poor, the classifier is likely to perform poorly on the reference images. The Reverse classifiers that are used in the study are Atlas forests (Zikic et al., 2014), the CNN model DeepMedic (Kamnitsas et al., 2017b) with a decreased amount of filters and consequently parameters to reduce overfitting to the single image input, and Atlas-based label propagation (Bai et al., 2013). It is important to note that the reference database used for evaluation can be the same as, or different from, the training database employed for training, cross-validation, and fine-tuning the original segmentation method. For measuring the segmentation accuracy, the maximum Dice score coefficient value that is found across all reference images is used. They found that Atlas Forests and, in particular, Single-Atlas label propagation yield accurate predictions (in terms of mean absolute error and correlation between predicted and true dice score coefficient) for different segmentation methods.

## 5.3  Drift Detection in the Medical Domain

Even though domain adaptation, which becomes relevant in case of dataset shift (domain shift), has gained significant attention over the last years (Guan & Liu, 2021), the methods on mere detection of data drift in the medical imaging context are still limited (Sahiner et al., 2023). This section will provide an overview of successfully applied drift detection methods for medical imaging workflows.

### 5.3.1  Classifier Based

Kore et al. (2024) ~~applied~~ apply dimensionality reduction using a trained Auto-Encoder (tAE) alongside Softmax outputs from a model trained on the source data, drawing parallels to the work of Rabanser et al. (2019). They ~~employed~~ employ their approach to detect data drift in chest X-ray images. The trained Auto-Encoder is a TorchXRayVision Auto-Encoder (Cohen et al., 2022) and the trained classifier is a TorchXRayVision model. They ~~applied~~ apply the MMD statistical test on the resulting dimensionally reduced embeddings to identify statistical differences between the source and target data. The shift detection methods ~~were~~ are tested on temporal imaging data, which initially constituted natural shifts (such as the introduction of COVID-19 in 2020, which caused a prevalence shift), and synthetically shifted data investigating specific population shifts. The shift factors studied ~~were~~ are institution, sex, patient age, patient class, and ICU admission status. The authors ~~found~~ find a correlation between the sensitivity of drift detection and the magnitude of the synthetic drift, with the tAE+trained classifier combination proving more sensitive than tAE alone, and the trained classifier being nearly as sensitive as the combined method.

Koch et al. (2024) ~~tackled~~ tackle the challenge of detecting clinically significant distribution shifts in retinal imaging data gathered during diabetic retinopathy screenings from multiple hospitals, each with diverse demographic populations. They ~~employed~~ employ a domain classifier to distinguish between the source and shifted datasets. They also ~~performed~~ perform MMD tests, where the kernel ~~was~~ is parametrised by a neural network, which ~~was~~ is trained separately on the source and shifted training sets. Additionally, they ~~analysed~~ analyse the softmax prediction outputs of a classifier trained to predict the $C$ classes from the source

dataset, using multiple univariate Kolmogorov-Smirnov tests. Similar to the approach taken by Kore et al. (2024), they ~~simulated~~ simulate subgroup shifts based on patient characteristics such as sex, ethnicity, image quality, and the presence of co-morbidities. For example, a shift in patient sex ~~was~~ is simulated by including only images from female patients in the target distribution. Their experiments ~~demonstrated~~ demonstrate that classifier-based tests consistently and significantly outperformed the other methods. However, crucial covariates that define relevant subgroup characteristics are often unmeasured or unidentified. Therefore by only controlling for a fraction of possible covariates, the shift might not ~~be originating~~ originate from the shifted feature, but from an altered composition of unobserved covariates.

### 5.3.2 Feature Based

The work of Stacke et al. (2020) ~~focused~~ focuses on quantifying the magnitude of domain shift by measuring shift in the learnt representation space of Histopathology imaging data. Under the assumption that in a well-trained model a convolutional layer will focus on image features that are relevant to the specific task and irrelevant features are discarded, Stacke et al. (2020) ~~developed~~ develop a metric (the representation shift metric $R_l$, which is defined for each layer $l$) that analyses the filter activations in each layer of a given CNN model. Even if the differences between the two domains appear minor in the image space, variations in training data statistics can lead to significant discrepancies in the internal model representations. First, all samples from the source and target dataset are passed through the model and for each layer $l$ and filter $k$ the feature maps $\varphi_{l,k}$ are averaged over the image dimensions $h$ and $w$:

$$c_{lk}(x) = \frac{1}{hw} \sum_{i,j}^{h,w} \varphi_{l,k}(x)_{i,j} \tag{8}$$

Next, the representation shift $R_l$ is derived by calculating the discrepancy between the distributions of $c_{lk}$ between the source and target dataset and then taking the average over all filters, which results in one $R_l$ per layer. In their study Stacke et al. (2020) ~~evaluated~~ evaluate three discrepancy metrics: Wasserstein distance, Kullback-Leibler divergence, and Kolmogorov-Smirnov statistic. If the data samples from the source and the target domains are statistically similar in the network's representation space, meaning a small representation shift $R_l$, then the target domain should be close to the source domain. Since the metric does not rely on annotated data, it can be used as a straightforward initial test to assess whether new data (*e.g.* histopathology images from a different scanner) is properly handled by an already trained model, *e.g.* whether the learned feature representation applies to the new data.

To increase the quality and quantity of medical data curation, Guo et al. (2023) propose MedShift as an automatic pipeline to combine datasets across different institutions. The approach assumes an available internal dataset which is treated as a baseline for evaluation of *shiftness* in external datasets. They demonstrate their method on two domains, namely musculoskeletal radiographs (MURA) with Stanford MURA (Rajpurkar et al., 2017) as external dataset, and chest X-rays with CheXpert (Irvin et al., 2019) and MIMIC (Johnson et al., 2019) as external datasets. Variational Auto-Encoders (VAE) are trained separately on each class of the internal dataset, thereby ~~,~~ learning a distributional representation which is queried and fed into a binary discriminator assessing whether the reconstructed query is part of the class distribution. Then the anomaly score is the sum of the reconstruction error of the VAE and the discriminator's prediction, with high scores indicating shift data. On the external dataset, the anomaly score is then calculated for each class and serves as the metric to construct an unsupervised class-wise clustering. Finally, the *shiftness* is quantified by evaluating a multi-class classifier trained on the internal dataset on the external dataset, by successively removing clusters with the highest anomaly score in each class, and monitoring the classifier's performance increase.

The proposed method is efficient to detect variability between datasets when they exhibit the same classes and could potentially be used to assess if a given dataset is similar to the training set used for model training. In their analysis, Guo et al. (2023) ~~found~~ find that higher anomaly groups are originating from variations in positioning, noise and image quality. Therefore, further investigation is needed to determine the specific types of shifts to which this method is sensitive.

### 5.3.3 Metadata Based

~~Additionally~~ In addition to image features, Merkow et al. (2023) also ~~integrated~~ integrate DICOM metadata (*e.g.* patient demographics, image formation metadata, image storage information) and predicted probabilities into an aggregated metric to detect temporal data drift in X-ray datasets using Rolling Detection Windows. To encode image appearance-based features and derive the latent space representation, they use a Variational Auto-Encoder (VAE), cf. Example 5.1. The statistical significance of the detected drift in continuous and categorical features is verified with Kolmogorov-Smirnov and Chi-Squared tests, respectively. Merkow et al. (2023) ~~simulated~~ simulate two drift scenarios: (1) performance degradation — randomly populating the stream of original data with hard data by only focusing on samples where the classifier exhibits low confidence for individual labels — and (2) clinical workflow failure — introducing lateral view images which the model ~~was~~ is not trained on; adding paediatric data that typical ML models are not authorised to report on. They ~~could~~ show that statistical significant variation in model inputs and outputs can be useful indicators of potential declines in model performance.

## 6  Discussion

2gray!25white

Table 1: Comparison of Bias Encoding Assessment, Data Drift Detection, Harmfulness Estimation, and Application in Medical Imaging Studies.

| Approach | Bias Assessment | | | Data Drift Detection | | | | | Harmfulness Evaluation |
| | Demographic Confounder | Non-Demographic Confounder | Synthetic Bias Datasets | Drift Detection | Prevalence Drift | Shift Detection | Synthetic Drift | Natural Drift | Segmentation Accuracy |
| --- | --- | --- | --- | --- | --- | --- | --- | --- | --- |
| Glocker et al. (2023a) | ✓ | | ✓ | | | | | | |
| Glocker et al. (2023b) | ✓ | | | | | | | | |
| Piçarra & Glocker (2023) | ✓ | | | | | | | | |
| Brown et al. (2023) | ✓ | | | | | | | | |
| Boland et al. (2024) | | ✓ | | | | | | | |
| Wachinger et al. (2019) | ✓ | ✓ | | | | | | | |
| Stanley et al. (2023) | | | ✓ | | | | | | |
| Stanley et al. (2024b) | | | ✓ | | | | | | |
| Kore et al. (2024) | | | | ✓ | | | ✓ | ✓ | |
| Koch et al. (2024) | | | | ✓ | | | ✓ | | |
| Merkow et al. (2023) | | | | ✓ | | | ✓ | | |
| Stacke et al. (2020) | | | | ✓ | | | ✓ | | |
| Roschewitz et al. (2023) | | | | ✓ | ✓ | | ✓ | | |
| Guo et al. (2023) | | | | | | ✓ | ✓ | | |
| Valindria et al. (2017) | | | | | | | | | ✓ |

In Sect. 4 and 5, we discuss methods to ensure reliable ML model deployment and maintain consistent prediction performance over time in clinical settings. Most of the methods presented have been tested on medical imaging data, while some, particularly those in Sect. 5.2 (drift detection), still need to be evaluated for this domain. To ease comparison between the discussed approaches, we have summarised their problem domain in Table 1.

The bias encoding assessment methods outlined in Sect. 4.1 can help practitioners gain transparency regarding bias encoding in pre-trained medical imaging models and to derive tailored bias mitigation strategies. Beyond assessing bias through output prediction disparities across sensitive subgroups (sex, race, age), examining the inner workings of bias encoding offers insights into how sensitive attributes are inter-related with

each other and related ~~with~~ to the primary task (disease detection). As Brown et al. (2023) and Glocker et al. (2023a) have demonstrated, encoding of sensitive attributes does not necessarily mean that those features are used for the primary task prediction. If bias is present, the impact of these sensitive attributes on model performance should be analysed (*e.g.* with the method proposed by Brown et al. (2023)) to distinguish between scenarios (a) and (b) as shown in Figure 4.

Further research is required to determine whether the findings of Glocker et al. (2023b) on bias encoding in chest radiography foundation models also apply to other foundation models. Given the high popularity of foundation models in the medical field (Azad et al., 2023) and the foundation models' use as a basis for other downstream task applications represents a key concern since any inherent biases of the foundation models might be inherited by all models that are fine-tuned on them (Bommasani et al., 2021). Therefore, it is crucial to tackle and alleviate biases in foundation models to guarantee fairness, inclusiveness, and ethical development within the medical domain (Azad et al., 2023).

Beyond demographic factors, works based on changing specific parameters (Lotter, 2024), or image modification (Boland et al., 2024; Sourget et al., 2025) ~~demonstrated~~ demonstrate the strong influence of non-demographic biases on model decisions. While some of the proposed methods can be used to analyse the influence of specifically selected attributes on model outcomes, there exists no widely adopted framework to detect general forms of shortcut learning in medical imaging to our knowledge. However, extending and improving synthetic bias generation frameworks such as SimBA (Stanley et al., 2023), so that they can accurately represent most real-world biases in the medical domain, ~~might~~ may be a promising step in this direction.

Determining the root causes of performance disparities remains challenging and will require future research. The presence of multiple potential sources of bias, including selection bias, annotation bias, and algorithmic bias — both demographic and non-demographic — makes it challenging to pinpoint the exact factors contributing to the disparities. Additionally, the interconnections between these biases complicate causal bias determination.

In addition to efforts aimed at mitigating bias in models trained on medical imaging data (Dinsdale et al., 2021; Correa et al., 2021; Yang et al., 2024b), it is crucial to collect a representative dataset to prevent the model from inheriting dataset bias. Attention should also be given to the labelling process, with procedures in place to resolve multi-annotator label disagreements, ensuring unbiased labels.

The methods mentioned previously are all specifically aimed at specific biases and require no stark modifications to the network. Some of the works from the same author, such as (Glocker et al., 2023a) and (Glocker et al., 2023b), build on the same framework. The methods do not contradict one another, as they operate within their own subtype of biases and with methods and metrics applied at distinct parts of the model inferences. A model that is already trained with a specific dataset can then be assessed successively and independently for the different kinds of possible biases.

When developing models, there is an inherent trade-off between tailoring a model to the specific deployment population — while risking bias from the under- or overrepresentation of certain subgroups — and creating a model based on a more representative dataset for the broader population (using *e.g.* reweighing strategies). The former approach may not generalise well to hospitals in other countries with different patient demographics, while the latter may lack the specificity needed for the deployment population, requiring fine-tuning for optimal performance.

Most of the drift detection methods discussed in Section 5.2 ~~were~~ are largely complementary to each other. Each method employs either layer activations, distributional representations, dimensionality reduction of the representations, or classifier outputs at different layers, obtaining drift scores when evaluated with a common metric such as the KS test. Ovadia et al. (2019) and Kore et al. (2024) emphasise how established uncertainty metrics do not detect shifts reliably at higher dimensions. Merkow et al. (2023) stress the importance of multimodality by utilising the metadata in addition to imaging data, in contrast to the other works used. Roschewitz et al. (2023) focuses on prevalence shift from, *e.g.* acquisition, which does not model time-dependent shifts like the other methods.

These methods were only evaluated on simulated data drifts. Thus, future research is necessary to assess these methods on real data drifts. However, this poses a challenge, as, given that we are mostly relying on temporal imaging data from a single source, it is inherently difficult to determine when a true statistically significant shift has occurred, which is essential for validating the effectiveness of these methods.

For implementing drift detection methods, it is crucial to set the intervals at which drift detection will be applied, thereby determining the appropriate number of images needed for the target dataset. Kore et al. (2024) ~~found~~ find that increasing the number of images in the target set improves sensitivity. However, this may not be feasible for smaller institutions or ML applications that handle relatively few cases, such as rare diseases. Thus, institutions may need to balance the frequency of drift detection with its sensitivity.

When a statistically significant data drift was detected and performance deterioration was estimated, expert labelling is necessary to annotate the data collected between the original model deployment and the drift detection ~~timepoint~~time point. This step is crucial to confirm whether there was an actual decline in performance. Then, retraining and re-validation of the model is necessary.

An intriguing area of study arises when drift is detected without corresponding performance deterioration. In such cases, analysing which covariates have changed (*e.g.* patient demographics, scanner type) through exploratory analysis can provide insights into which populations the model remains generalizable to. However, this analysis must be approached with caution, as, even if generalizability to certain populations is observed, there is no guarantee that the model is generalizable to populations with similar distributions, since unknown confounding factors may still be present.

## 6.1 Mitigation Strategies

In this work, we primarily focus on methods for *detecting* bias and data ~~drifts, leaving the~~ drift, leaving an in-depth discussion of automated mitigation strategies for future exploration. ~~Alternatives to re-labelling, re-training,~~ Nevertheless, we briefly outline the main approaches, clarifying which address data bias or data drift. To make the connection between lifecycle stages and deployment context explicit, we provide a comprehensive crosswalk in Table 2. This table maps each stage of the ML lifecycle to either Design-time Robustness (DG) or Post-deployment Adjustment (DA), highlighting the intended mitigation focus, typical techniques, and relevant medical imaging references.

Pre-deployment techniques can be applied during data acquisition, curation, splitting, and preprocessing (Tejani et al., 2024b; Drukker et al., 2023b; Banerjee et al., 2023), aiming to reduce imbalances across sensitive attributes and acquisition settings. During model development, established methods for reducing dependence on spurious correlations include IRM (Arjovsky et al., 2020), Group-DRO (Sagawa* et al., 2020), and resampling (Idrissi et al., 2022). In recent years, model scaling and large-scale pre-training (foundation models) have emerged as strong baselines for OOD robustness, often outperforming specialised mitigation algorithms in counteracting drift and improving generalisation (Oquab et al., 2024; Siméoni et al., 2025). This trend is further supported by recent evidence showing that careful selection among pre-trained models can itself yield substantial gains in OOD generalisation and calibration performance (Naganuma et al., 2025). Their efficacy, however, should be verified on benchmark datasets such as WILDS (Koh et al., 2021) and DomainBed (Gulrajani & Lopez-Paz, 2021), which capture both distributional and subgroup shifts relevant to bias and drift. Notably, these models can still fail when pre-training data are poorly aligned with downstream domains, underscoring their sensitivity to data–domain mismatch and clarifying that scaling alone is not a universal solution for domain generalisation or drift resilience (Teterwak et al., 2025). Moreover, subgroup definition remains a critical factor: improving fairness with respect to one set of subgroups may require defining mitigation subgroups differently, underscoring the importance of careful subgroup specification in bias-mitigation settings (Alloula et al., 2025).

In contrast, methods such as Domain Adaptation (DA) (Kamnitsas et al., 2017a; Ouyang et al., 2019; 2022b;a; Guan & Liu, 2 and continuous retraining (Kore et al., 2024; Koch et al., 2024) are primarily aimed at neutralising the temporal evolution captured by data drift. DA aligns the model's input or feature space with the deployment distribution, directly counteracting shifts such as acquisition-related or other covariate drift, while concept drift typically requires adaptation of the predictive mechanism itself via fine-tuning or retraining.

Table 2: Crosswalk mapping lifecycle stages to Design-time Robustness (DG) and Post-deployment Adjustment (DA), showing whether methods primarily address data bias or data drift, with representative techniques and evaluation considerations.

| Lifecycle Stage | Intent | DG / DA | Bias / Drift | Typical Techniques | Mitigation and Evaluation Considerations |
|---|---|---|---|---|---|
| Data acquisition & curation (Tejani et al., 2024b; Drukker et al., 2023b; Banerjee et al., 2023) | Reduce sampling imbalance | DG | Bias (also reduces drift risk) | Stratified sampling, over-/under-sampling, balanced cohort design, metadata-aware collection | Prevents subgroup under-representation; evaluated via subgroup prevalence and performance gaps |
| Pre-processing & augmentation (Tejani et al., 2024b; Banerjee et al., 2023) | Increase robustness to known variation | DG | Bias + acquisition drift | Domain-aware augmentation, style transfer, harmonisation, normalisation | Encourages invariance to site or scanner differences; assessed via held-out domain tests and subgroup comparison |
| Model training (Sagawa* et al., 2020; Idrissi et al., 2022; Alloula et al., 2025; Naganuma et al., 2025) | Reduce reliance on spurious correlations | DG | Bias | Group-DRO, resampling, model scaling and large-scale pre-training (foundation models) | Careful subgroup definition; larger/foundation models can mitigate drift impact and improve generalisation (not universally) |
| Domain Generalisation (Kamnitsas et al., 2017a; Ouyang et al., 2019; 2022b;a; Guan & Liu, 2021; Chen et al., 2019) | Generalise to unseen deployment settings | DG | Covariate / acquisition drift | Multi-source DG, domain-invariant representation learning, meta-learning | Validated through OOD generalisation gaps and worst-domain accuracy |
| Validation & stress testing (Stacke et al., 2020; Koh et al., 2021; Bercea et al., 2025; Gutbrod et al., 2025) | Assess pre-deployment generalisation to new domains | DG | Bias and Drift | Shift/OOD benchmark datasets (e.g., Camelyon17-WILDS), stress tests | Evaluate robustness and fairness across demographic, institutional, and acquisition shifts; quantify design-time robustness across subgroup disparities and distributional shifts |
| OOD detection (Yang et al., 2024a; Zimmerer et al., 2022; Tan et al., 2021; Schlüter et al., 2022; Naval Marimont et al., 2024) | Flag inputs outside validated manifold | DG (trained) + DA (operational) | Drift (also rare subgroup signals) | Mahalanobis / confidence-based detectors, density estimators, deep-feature OOD scoring | Enables selective prediction or adaptation triggers; evaluated via AUROC and coverage–error trade-offs |
| Monitoring & drift detection (Sahiner et al., 2023; Merkow et al., 2023) | Identify deployment shifts | DA | Covariate & concept drift | Population-statistics monitors, divergence tests, label-delay-aware monitoring | Triggers retraining or human oversight; evaluated via detection delay, false alarms and subgroup sensitivity |
| Domain Adaptation (DA) (Kamnitsas et al., 2017a; Ouyang et al., 2019; 2022b) | Adapt model to deployment data | DA | Covariate drift (limited concept drift) | Unsupervised/semi-supervised DA, feature alignment, self-training | Restores alignment with deployment distribution; measured via post-adaptation accuracy and subgroup stability |
| Continuous retraining (Kore et al., 2024; Koch et al., 2024) | Maintain relevance over time | DA | Concept drift | Incremental fine-tuning, periodic retraining, online learning | Maintains predictive validity; requires labelled pipelines and longitudinal subgroup tracking |
| Operational controls (Guo et al., 2023; Merkow et al., 2023) | Safety under uncertainty | DA (runtime) | Residual bias & drift | Selective prediction, clinician referral, conservative failover | Ensures patient safety; evaluated via coverage vs reliability and human–AI joint outcomes |

Out-of-distribution (OOD) detection approaches (Yang et al., 2024a; Zimmerer et al., 2022; Tan et al., 2021; Schlüter et al., serve as crucial mitigation enablers by identifying samples that deviate significantly from the validated training manifold. This allows safe operational choices such as selective prediction, human review, or triggering targeted adaptation pipelines. Similarly, monitoring and drift detection pipelines (Sahiner et al., 2023; Merkow et al., 2023) have been applied in medical imaging to identify temporal or subgroup shifts in X-ray datasets and other modalities, supporting decisions for continuous retraining or operational safeguards. Table 2 explicitly summarises these strategies across the model lifecycle, showing how DG methods support pre-deployment robustness (*e.g.* data curation, augmentation, robust training including model scaling and foundation models, domain generalisation, validation and stress testing with shift/OOD datasets, *e.g.*, WILDS, DomainBed, NOVA, OpenMIBOOD (Stacke et al., 2020; Koh et al., 2021; Gulrajani & Lopez-Paz, 2021; Bercea et al., 2025; Gutbrod et al., 2025) while DA methods support post-deployment adjustment (*e.g.* OOD detection methods, drift monitoring, domain adaptation, continuous retraining, operational controls (Guo et al., 2023; Kore et al., 2024; Koch et al., 2024; Merkow et al., 2023). This mapping clarifies which methods are relevant for design-time versus operational reliability, making the workflow, novelty, and contribution of the survey explicit to the reader.

### 6.2 Link between Design-time Robustness and Post-deployment Adjustment

Understanding the links between design-time generalisation (DG) and ~~re-evaluation are not covered in depth. Nevertheless, we highlight two promising directions that merit further exploration in the literature: (a) domain adaptation techniques (see, e.g., (Kamnitsas et al., 2017a; Ouyang et al., 2019; 2022b;a; Guan & Liu, 2021; Chen et al., 2019)), which modify inputs to align them with a target distribution where a trained model remains effective,~~ post-deployment adaptation (DA) is critical for guiding future work on robust medical imaging AI. Identifying underexplored combinations and gaps can inform the development of lifecycle-aware models that maintain performance across diverse domains and evolving clinical environments. We conducted a structured review of peer-reviewed studies on medical imaging AI published from 2021 to July 2025. Searches were performed in IEEE Xplore, Scopus, and Google Scholar using keywords related to data shift, bias, DG and DA both with their lifecycle stages, explicitly in medical imaging. Studies were included if they explicitly addressed methods that link design-time generalisation and post-deployment adaptation. For each study, we extracted the lifecycle stage targeted, see Tab. 2, the type of distributional shift addressed, and the methodological connection between DG and DA.

A primary form of linkage is DG-informed pre-training that facilitates DA. Variational autoencoders encode anatomical priors during DG-style training, guiding adaptation to new target domains (Yao et al., 2022). Frequency- and ~~(b) out-of-distribution detection approaches (see, e.g., (Yang et al., 2024a; Zimmerer et al., 2022; Tan et al., 2021; Schlüter et al., 2022; Naval Marimont et al., 2024; Baugh et al., ))~~ spatial-domain transfer under multi-teacher distillation enables DA to leverage robust pre-trained features without adversarial training (Liu et al., 2023b). Self-supervised 3D anatomical landmark extraction learns domain-invariant features during DG that support large cross-modality adaptation (Cui et al., 2021). Early feature pseudo-labelling bridges DG pre-processing with DA adaptation (Sheikh & Schultz, 2022).

Some studies conceptualise DG and DA as a continuum rather than separate stages. Hybrid frameworks perform DG-style training on multiple source domains, then apply DA at test time for new target domains. For instance, (Weihsbach et al., 2025) implement domain-generalised pre-training followed by test-time adaptation, significantly improving segmentation performance across CT-MRI, abdominal, spine, and cardiac imaging. Adaptive entropy regularisation (Shi & Feng, 2023), dual adversarial attention (Chen et al., 2022), conditional diffusion models (Zhao et al., 2024), and visual prompting (Lu et al., 2025) integrate DG-trained features into DA pipelines, highlighting the practical continuity between the two stages.

Out-of-distribution (OOD) robustness is a key mechanism connecting DG and DA. DG strategies aim to improve model generalisation to unseen domains using domain-invariant representations, multi-source training, and stress-testing on held-out sites (Guan & Liu, 2021; Yao et al., 2022; Sheikh & Schultz, 2022). Simple preprocessing modifications, such as window-based normalisation using local image statistics, can enhance OOD generalisation in medical images (Zhou et al., 2024). These DG techniques reduce sensitivity

to acquisition and scanner variations, improving the effectiveness of downstream DA. DA methods build on DG-derived OOD robustness, since models that have been pre-trained to withstand unseen distribution shifts provide a stronger starting point for post-deployment alignment, reducing adaptation failures and improving performance in evolving clinical environments. Unsupervised DA can fail if source and target domains differ greatly in contrast, spatial structure, or texture (van Tulder & de Bruijne, 2023). Pre-deployment DG strategies, like style-consistency or structural priors, increase adaptation success in such challenging OOD scenarios (Chen et al., 2024; Iacono & Khan, 2024). Hybrid pipelines exploit this relationship: DG pre-training provides OOD-invariant features, which DA then fine-tunes on unlabelled target domains, creating a cycle of robustness-to-adaptation.

Privacy- and resource-aware methods further extend DG–DA links. Source-free DA adapts models without access to source data while leveraging DG-trained features (Wu et al., 2025). Pseudo-labelling and mutual-information maximisation refine DA predictions using DG pre-training (En & Guo, 2024; Kumari & Singh, 2025). Cross-population adaptation in chest X-rays shows that DG captures demographic variability, while DA fine-tunes for new populations (Musa et al., 2025). Image-to-image translation and cross-modality synthesis reduce annotation burden and enable hybrid DG–DA pipelines (Tomar et al., 2021; Li et al., 2022).

While these links are increasingly explored, several combinations remain understudied. Few studies examine DG-informed DA in structured post-detection interventions, such as selective prediction or automated model recalibration after detecting OOD inputs. Similarly, there is limited evidence on continuous DG–DA loops, where models iteratively refine DG representations based on successive DA feedback from evolving target domains. The intersection of DG with privacy-preserving DA, particularly source-free adaptation combined with OOD-robust DG, is only beginning to be explored. Another under-covered area is DG–DA across multiple shifts simultaneously, such as adapting to both cross-modality and cross-population variations while maintaining generalisation to unseen scanners or imaging protocols.

Additionally, most hybrid approaches are confined to segmentation or modality-transfer tasks. Few studies explore DG–DA links in diagnostic classification, time-series imaging, or multimodal data, leaving a gap in applying these strategies beyond typical computer vision settings. Finally, while OOD robustness is recognised as a key link, empirical evidence connecting specific DG methods (e.g., style-consistency, ~~often coupled with biased model selection, where models tailored to specific populations are chosen based on explicit or implicit patient class parameters.~~

~~The first avenue has already demonstrated encouraging results in transferring tasks across modalities (Ouyang et al., 2022b), while the second offers a practical pathway for production deployment, though it requires careful validation for each sub-population. Readers interested in mitigation strategies beyond detection may find these works particularly valuable as starting points.~~ multi-source training) to DA performance under varying OOD conditions remains sparse. These gaps highlight opportunities for end-to-end DG–DA frameworks that systematically integrate pre-deployment robustness, continuous adaptation, OOD detection, and post-deployment adjustment.

Despite these advances, the literature still lacks comprehensive solutions linking design-time generalisation, OOD robustness, and post-deployment adaptation across the full ML lifecycle. Filling these gaps is essential for building flexible, robust, and clinically deployable medical imaging AI systems.

# 7 Conclusion

The safe integration of machine learning into clinical practice requires robust systems that are reliable and effective. The ideal of total fairness remains elusive, as group-level statistical parity can conflict with individual-level accuracy and clinically relevant correlations. Therefore, the practical goal is to make model biases tractable and interpretable.

In this survey, we present a unified framework for addressing two critical threats to ~~this~~ reliability: unexamined bias encoded during development and performance degradation from data drift post-deployment. ~~Our work provides a critical overview of the state-of-the-art methods needed to identify these challenges, even when ground truth is unavailable. This pursuit of transparency is not merely a technical exercise but an~~

~~ethical and regulatory imperative. Emerging regulations, such as the EU AI Act (European Council, 2023) and California's AI legislation (California State Legislature, 2023b;a), mandate a shift from static validation to continuous life-cycle governance. These frameworks demand the very tractability this survey champions: the ability to monitor performance , understand potential discrimination risks,~~ By explicitly mapping lifecycle stages to *Design-time Robustness (DG)* and ~~make informed decisions to ensure patient safety and build clinical trust (European Commission, 2021). The methods we review are foundational to meeting these requirements.~~ *Post-deployment Adjustment (DA)* strategies in Tab. 2, we provide an evidence-based framework for jointly evaluating bias and drift, and for guiding robust model deployment in clinical settings. The crosswalk shows how existing methods mitigate bias, drift, or both jointly, and clarifies how modern approaches, including model scaling and large-scale pre-training, fit within the lifecycle, while also acknowledging that foundation models can still fail when pre-training data are misaligned with downstream populations.

Our literature review highlights several key findings on the links between DG and DA strategies. First, DG-informed pre-training and OOD robustness are central enablers for effective DA, providing a foundation for models to adapt to new clinical domains and populations. Second, hybrid DG–DA frameworks are increasingly explored, particularly in segmentation and modality-transfer tasks, but remain limited in diagnostic classification, multimodal imaging, and continuous adaptation loops. Third, many underexplored combinations remain, such as structured post-detection interventions, source-free privacy-preserving DA, and multi-shift adaptation, emphasising the need for end-to-end lifecycle-aware solutions. Finally, the interplay between DG and DA demonstrates that robustness and fairness cannot be assessed in isolation: pre-deployment interventions must be evaluated alongside post-deployment adaptation to ensure both bias mitigation and performance stability over time.

This framing makes clear that neither bias nor drift can be reasoned about in isolation: robustness claims made at design time may not hold under evolving deployment conditions, and fairness assessments are incomplete without accounting for population or acquisition shifts over time. Our integrated view therefore highlights not only current methodological strengths, but also open challenges where bias mitigation and drift resilience still need to be co-developed and benchmarked together in realistic medical settings.

By unifying the concepts of bias assessment and drift detection, our survey provides an essential roadmap for the medical AI community. It equips researchers and clinicians with ~~the integrated perspective needed to move beyond the pursuit of perfect fairness and toward~~ an integrated perspective to understand and manage both sources of unreliability. The key take-home message is that biases and drift are interrelated threats: by detecting, characterising, and monitoring them jointly, practitioners can make informed decisions to ensure patient safety, maintain model performance over time, and guide the development of ~~resilient~~ robust, fair, and clinically reliable AI systems. ~~The goal is to create tools where biases are not just hidden liabilities but are identified, understood, and managed.~~

This approach is not merely technical but also aligns with emerging regulatory frameworks, such as the EU AI Act (European Council, 2023) and California's AI legislation (California State Legislature, 2023b;a), which mandate continuous lifecycle governance, monitoring of performance, and proactive management of potential discrimination risks (European Commission, 2021). The methods we review provide the foundational tools to meet these requirements, demonstrating how the survey advances the state-of-the-art by connecting bias and drift mitigation in a single, integrated framework.

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
