# OpenReview forum: "Bias Assessment and Data Drift Detection in Medical Image Analysis: A Survey"
_TMLR — Rejected by TMLR_

### Review · Reviewer_6Sbw · 2025-10-05

**Summary Of Contributions:**

**Summary**

This survey examines bias (pre-deployment subgroup disparities) and data drift (post-deployment distributional change) as the two main reliability threats in medical-imaging ML systems. The authors structure their taxonomy along the ML lifecycle (bias assessment during model development and drift assessment during deployment) and provide definitions, causes, and representative methods for each. Using accessible language and clinical examples, they review fairness metrics, shortcut learning, and confounder detection, as well as covariate, label, and concept shift, culminating in a synthesis table (Table 1). The work explicitly targets collaboration between technical and clinical communities.

**Strengths and Weaknesses**

**Strengths**

- **Practical lifecycle framing.** Figure 1 (p. 3) presents a clear visual taxonomy linking bias (acquisition, selection, annotation, algorithmic) and drift (covariate, prior, concept). This operational split helps clinicians reason about reliability checkpoints.
- **Accessible exposition.** The paper translates complex fairness criteria (independence, separation, sufficiency) and drift types into clinically interpretable terms, supporting its stated goal of bridging communities.
- **Methodological breadth.** Sections 4–5 synthesize a wide range of assessment techniques (from PCA-based bias encoding to Autoencoder-based drift detection) and anchor them in medical-imaging use cases.

**Weaknesses**

1. **Boundary between “bias” and “drift” is mainly operational.**
   The authors’ distinction like bias as static pre-deployment and drift as dynamic post-deployment, is practical but not theoretically solid. They themselves note that "sample selection bias causing dataset shift" (Sec. 3, p. 9) and describe acquisition bias using the term domain shift (Sec. 2.1.1). Thus, the boundary rests more on the time of occurrence than on a clear statistical separability. Formalizing each concept through the factorization and the causal/anti-causal settings (Sec. 3.1) would strengthen the framework’s rigor.

2. **Partial alignment with Domain Generalization (DG) / Domain Adaptation (DA) literature.**
   Section 6 briefly references DA and OOD detection, yet the paper stops short of mapping its lifecycle taxonomy to DG (design-time robustness) and DA (post-deployment adjustment). Since these are established terms for pre- vs. post-deployment reliability, an explicit crosswalk table would clarify novelty and connect the survey to the broader ML discourse.

3. **Disconnection from Foundational OOD and Calibration Research.** This survey completely overlooks the most significant advances in recent OOD generalization research, particularly the role of model scaling and pre-training. Modern large-scale pre-trained models have been shown to improve OOD generalization and calibration performance as they scale, often without explicit regularization [1]. This phenomenon directly addresses the paper's themes: scaling helps models learn more invariant features, mitigating "bias" and thereby improving robustness against future drift. The omission of this entire line of research is a major flaw. Furthermore, the paper fails to cite or discuss foundational benchmarks like DomainBed [2], WILDS [3], or Uncertainty Baselines [4]. These resources are essential for practitioners seeking to evaluate and select methods for their own problems.

4. **Mitigation perspective remains underdeveloped.**
   Section 6 acknowledges but defers mitigation strategies (IRM [5], DRO [6], DA, OOD detection). A compact section linking each method to the lifecycle in Figure 1 (e.g., bias-oriented vs. drift-oriented techniques) would increase the survey’s practical impact without bloating length.

5. **Citation and presentation inconsistencies.**
   There are instances in the paper where `\cite{}` and `\citep{}` are used inconsistently, which negatively affects readability.

6. **Poor Structure.** For a survey that exceeds 20 pages, the introduction lacks a roadmap outlining the paper's structure. This makes the document difficult to navigate and hinders reader comprehension.


```
[1] An Empirical Study of Pre-trained Model Selection for Out-of-Distribution Generalization and Calibration, TMLR2025
[2] In Search of Lost Domain Generalization, ICLR2021
[3] WILDS: A Benchmark of in-the-Wild Distribution Shifts, ICML2021
[4] Can You Trust Your Model's Uncertainty? Evaluating Predictive Uncertainty Under Dataset Shift?, NeurIPS2019
[5] The Risks of Invariant Risk Minimization, ICLR2021
[6] Distributionally Robust Neural Networks, ICLR2020
```

**Additional Comments:**

This paper tackles a vital topic with ambition. However, its core conceptual contribution is underdeveloped and isolated from the broader conversation in the ML research community. The requested revisions are substantial but necessary. If the authors can successfully integrate their survey with the foundational work on OOD generalization and clarify their framework's novelty and utility, the paper could become a significant contribution.

**Audience:**

Yes

**Audience Explanation:**

The topic of model reliability, fairness, and robustness in medical AI is of high interest to a broad segment of the TMLR audience, including practitioners and researchers in healthcare, ML safety, and trustworthy AI. The paper serves as a useful, albeit flawed, introduction to the unique challenges of distribution shifts in the medical domain. If revised to address its significant weaknesses, it has the potential to be a valuable resource.

**Broader Impact Concerns:**

None. The paper aims to improve the safety and fairness of medical AI, which is a positive societal goal.

**Claims And Evidence:**

No

**Claims Explanation:**

The central claim—that a clear distinction between "bias" and "data drift" offers a useful framework for ensuring model reliability—is not convincingly supported.
The submission provides no evidence, either theoretical or empirical, to justify why its novel framework is superior to or provides new insights beyond these existing concepts.

Second, the survey omits a vast body of crucial evidence from recent OOD generalization research. It does not discuss the well-documented finding that scaling up pre-trained models improves robustness to distribution shifts and enhances calibration, a phenomenon that directly bridges the paper's concepts of "bias" (mitigated through learning more invariant features from diverse data) and "drift" (improved robustness against it). By ignoring seminal benchmarks like DomainBed and WILDS, the paper also fails to ground its discussion in the evidence generated by the broader ML community on the efficacy of different algorithms.

**Requested Changes:**

1. **Formalize the bias–drift boundary.**
   Introduce a subsection that derives each concept from \(P(X,Y)\) decomposition and causal/anti-causal perspectives, clarifying observables and interaction cases (e.g., selection bias leading to covariate shift).

2. **Map the framework to DG / DA / OOD terminology.**
   Add a compact table aligning *Bias Assessment → DG-style robustness methods* and *Drift Assessment → DA/OOD monitoring*, specifying which terms of \(P(X,Y)\) are affected.

2.  **Incorporate Recent OOD and Calibration Research.** The paper must be updated to include a discussion on the impact of model scaling and large-scale pre-training on OOD generalization and calibration. This discussion should be situated within the context of foundation models. Citing recent empirical studies like [1] is crucial to provide practitioners with actionable insights on pre-trained model selection. The survey must also introduce key benchmarks like DomainBed[2], WILDS[3], and Uncertainty Baselines[4], and discuss their findings.

3.  **Expand on Mitigation Strategies.** A chapter or section dedicated to mitigation strategies is required. This section should review major approaches like IRM[5], DRO[6], and DA, analyzing them through the lens of the paper's framework to show whether they address "bias" or "drift".

5. **Standardize citation and style.**

6. **Add a unified crosswalk figure/table.**
   Place a one-page summary (at the end of the introduction or start of Sec. 2) aligning taxonomy ↔ DG/DA/OOD ↔ \(P(X,Y)\) components ↔ clinical examples ↔ recommended methods.


```
[1] An Empirical Study of Pre-trained Model Selection for Out-of-Distribution Generalization and Calibration, TMLR2025
[2] In Search of Lost Domain Generalization, ICLR2021
[3] WILDS: A Benchmark of in-the-Wild Distribution Shifts, ICML2021
[4] Can You Trust Your Model's Uncertainty? Evaluating Predictive Uncertainty Under Dataset Shift?, NeurIPS2019
[5] The Risks of Invariant Risk Minimization, ICLR2021
[6] Distributionally Robust Neural Networks, ICLR2020
```

---

> ### Author Response · Authors · 2025-11-25
> **Answer 6SBw**
>
> We sincerely thank the reviewer for the thorough and constructive feedback, the extensive set of references, and the clear guidance on how to strengthen the conceptual and technical contributions of our survey. We greatly appreciate the time and care invested in this review, and we address the requested changes point by point below.
>
> ## Regarding the bias-drift boundary:
> While the temporal distinction between bias (pre-deployment) and drift (post-deployment) may appear trivial, we have now defined it more formally. Furthermore, as suggested, we have introduced causal and anti-causal perspectives with examples.
>
> ## Regarding Mapping the Framework to DG/DA/OOD:
> We thank the reviewer for highlighting the partial alignment with the Domain Generalization (DG) and Domain Adaptation (DA) literature. To address this, we have introduced Tab.2, which explicitly maps each stage of the ML lifecycle to either Design-time Robustness (DG) or Post-deployment Adjustment (DA). The table highlights the intended mitigation focus, typical techniques, bias vs. drift addressed, and relevant medical imaging references. In addition, Section 6 has been revised to explicitly discuss the DG vs. DA distinction, linking pre-deployment strategies (e.g., robust model training, data curation, validation and stress testing) to DG and post-deployment strategies (e.g., OOD detection, drift monitoring, continuous retraining, operational controls) to DA. By making this lifecycle mapping explicit, we clarify the novelty of our survey in connecting classical reliability terms from the broader ML literature to practical mitigation strategies in medical imaging. We believe these changes fully address the concern and situate our survey within the DG/DA discourse, making the lifecycle taxonomy and its implications for pre- vs. post-deployment reliability immediately clear to the reader.
>
>
>
> ## Regarding Disconnect to Recent OOD and Calibration Research
> We would like to be careful with the direct, causal beneficial claims that these SOTA foundational methods lead to guaranteed bias and drift detection performances. Works like DomainBed and Uncertainty SOTAs do not explicitly claim, both quantitatively and qualitatively, on tackling the bias and drift problems that we addressed here, rather the specific calibration, domain adaptation, and OOD performance. As such, while intuitively they should lead to improved bias and drift detection performances, we could not find any direct mathematical proofs that guarantee such causal beneficial relationships. Additionally, as specified in our paper, the works done by Ovadia et al. [1] and Kore et al.  [2] explicitly show that uncertainty metrics such as calibration and entropy are not reliable indicators to shifts. Works on uncertainty SOTAs are inherently focused on improving model calibration and uncertainty scores.
> In summary, we believe that an in-depth discussion of uncertainty and general OOD methods would detract from the focus on state-of-the-art bias and drift detection and mitigation. For this reason, we omit it.
>
> [1] Ovadia, Yaniv, et al. "Can you trust your model's uncertainty? evaluating predictive uncertainty under dataset shift." Advances in neural information processing systems 32 (2019).
>
> [2] Kore, Ali, et al. "Empirical data drift detection experiments on real-world medical imaging data." Nature communications 15.1 (2024): 1887.
>
>
>
> ## Regarding the potential to expand on Mitigation Strategies:
> We thank the reviewer for pointing this out. While a comprehensive analysis of mitigation strategies goes beyond the main scope of our survey, we fully agree that an overview of these strategies would strengthen our contribution. Therefore, in the revised manuscript, we expanded Section 6 by adding a dedicated subsection including an outlook on mitigation strategies (now Sect. 6.1).
>
> ## Regarding comments about adding a unified crosswalk table:
> We thank the reviewer for highlighting the importance of a clear and accessible roadmap for a survey of this length. In response, we have expanded the existing roadmap at the end of the introduction to more clearly describe the content and purpose of each section, including: (i) the types of bias reviewed, (ii) the types of data drift considered, (iii) the corresponding assessment challenges, and (iv) how these concepts relate to the methods discussed later in the survey. The updated roadmap now gives a more accessible, section-by-section overview of the taxonomy used throughout the paper and improves the reader’s ability to follow the survey structure.
>
> While a full one-page summary was beyond the scope of our introduction, we believe the revised roadmap provides substantially improved navigational clarity.
>
>
>
> **Finally, we have carefully revised the manuscript to ensure consistent citation and style.**

---

> > ### Comment · Reviewer_6Sbw · 2025-11-25
> > **Contradiction regarding Mitigation Scope and OOD Generalization Context**
> >
> > I thank the authors for their detailed response and for revising the manuscript. The clarifications on the bias-drift boundary and the inclusion of Table 2 have strengthened the paper.
> >
> > However, I maintain that excluding the discussion on recent OOD generalization research (foundation models, scaling effects) and key benchmarks (WILDS, DomainBed) creates a disconnect between the survey's scope and the current state of the field.
> > The authors argue that including these topics would "detract from the focus on state-of-the-art bias and drift detection and mitigation," as these works do not explicitly claim to tackle "detection." I find this reasoning inconsistent with the revised manuscript for the following reasons:
> >
> > 1. Inconsistency with Section 6.1 (Mitigation):
> > The authors have rightly added Section 6.1 and Table 2, explicitly mapping lifecycle stages to Design-time Robustness (Domain Generalization). The paper now cites specific DG methods like IRM and Group-DRO. It is therefore contradictory to include these methods while excluding the standard benchmarks (like WILDS and DomainBed) used by the community to evaluate OOD generalization and robustness against distribution shifts (drift) and sub-population shifts (bias). Discussing DG techniques without referencing the benchmarks that reveal their efficacy (or lack thereof) renders the survey incomplete.
> >
> > 2. Mitigation via Scaling:
> > The survey aims to guide practitioners on "building fair... and reliable ML systems" (Abstract). A critical modern finding is that model scaling and large-scale pre-training (foundation models) often provide stronger baselines for OOD robustness (mitigation of drift impact) than specialized algorithmic interventions. By omitting this, the survey risks recommending older techniques without acknowledging that simple scaling might be a more effective mitigation strategy in 2025.
> >
> > 3. Scope of Reliability:
> > While calibration metrics might not be perfect detectors of drift (as the authors noted referencing Ovadia et al.), the broader OOD research provides essential context on how models behave under drift. Excluding this literature narrows the scope of "Reliability" artificially, ignoring the "Robustness" aspect the authors themselves highlight in the introduction.
> >
> > I strongly urge the authors to reconsider. I am asking for a balanced view: since the paper now explicitly covers Mitigation and Design-time Robustness, it must briefly acknowledge the dominant role of foundation models/scaling in modern robustness and cite the benchmarks that serve as the standard evaluation grounds for the very shifts (bias/drift) this survey discusses.

---

> > > ### Author Response · Authors · 2025-11-25
> > > **Response on OOD Generalisation, Scaling, and Benchmark Integration**
> > >
> > > We thank the reviewer for their prompt and considered feedback. The manuscript has been revised to address the concerns as follows.
> > >
> > > 1. **Inconsistency with Section 6.1 (Mitigation)**: We have explicitly integrated standard OOD generalisation benchmarks, including WILDS, DomainBed, NOVA, and OpenMIBOOD, into Section 6.1 and Tab. 2 under “Validation & Stress Testing.” This ensures that the discussion of DG methods (e.g., IRM, Group-DRO) is aligned with the empirical frameworks used to evaluate performance under both subgroup shifts (bias) and distributional shifts (drift).
> > >
> > > 2. **Mitigation via Scaling**: The text now acknowledges that model scaling and large-scale pre-training (foundation models) often provide stronger baselines for OOD robustness and mitigation of drift impact than specialised algorithmic interventions. We also highlight evidence that such models can fail when pre-training data are poorly aligned with downstream domains [1], providing a balanced perspective.
> > >
> > > 3. **Scope of Reliability**: We clarify that reliability encompasses both design-time robustness and post-deployment adjustment. While calibration metrics may not perfectly detect drift, OOD generalisation research provides essential context on model behaviour under distributional shift. These considerations are now integrated into Section 6.1, Tab. 2, and the conclusion, ensuring a consistent interpretation of reliability throughout the manuscript.
> > > Overall, the revised manuscript now presents a balanced, evidence-based view of DG and DA methods, scaling effects, and standard benchmarks, addressing both bias and drift jointly.
> > >
> > > [1] Teterwak, P., Saito, K., Tsiligkaridis, T., Plummer, B. A., & Saenko, K. (2024). Is Large-Scale Pretraining the Secret to Good Domain Generalization?. arXiv preprint arXiv:2412.02856. [ICLR 2025]

---

> > > > ### Comment · Reviewer_6Sbw · 2025-11-25
> > > > **Thank you for the revision: Request for one final citation for balanced discussion**
> > > >
> > > > I appreciate the authors' responsiveness and the substantial improvements made to the manuscript. The inclusion of Section 6.1 and the updated Table 2 explicitly connecting DG/DA methods, scaling, and standard benchmarks (WILDS/DomainBed) has resolved my major concerns regarding the scope and currency of the survey.
> > > > I particularly appreciate the inclusion of Teterwak et al. (2025) to highlight the limitations of scaling when alignment is poor. However, regarding the statement: "In recent years, model scaling and large-scale pre-training (foundation models) have emerged as strong baselines for OOD robustness..."
> > > >
> > > > The manuscript currently cites Oquab et al. (2024) here. To make this statement empirically robust within the specific context of OOD generalization and Calibration benchmarks (which this survey focuses on), it is crucial to cite the evidence of scaling laws specifically on DomainBed/WILDS.
> > > > The paper I mentioned in my previous review, "[1] An Empirical Study of Pre-trained Model Selection for Out-of-Distribution Generalization and Calibration (TMLR 2025)," provides the direct quantitative evidence that larger models and datasets improve performance and calibration on DomainBed/WILDS.
> > > > Citing [1] alongside Teterwak et al. (2025) would provide a complete and balanced dialectic: [1] establishes scaling as a strong baseline on these benchmarks, while Teterwak et al. nuances this by showing its dependency on alignment.With this final addition to Section 6.1 to ground the "positive" aspect of scaling in the relevant literature, I am confident to recommend acceptance.
> > > >
> > > > [1] Naganuma, H., et al. "An Empirical Study of Pre-trained Model Selection for Out-of-Distribution Generalization and Calibration." Transactions on Machine Learning Research (2025).

---

> > > > > ### Author Response · Authors · 2025-11-26
> > > > > **Response to on Scaling Evidence and Section 6.1 Revisions**
> > > > >
> > > > > We thank the reviewer for the prompt follow-up, and we appreciate the reminder to integrate this source, which is indeed well-justified. In Section 6.1, we have now incorporated Naganuma et al. (2025) exactly as recommended to provide direct, benchmark-specific evidence that larger pre-trained models improve OOD generalisation and calibration on DomainBed and WILDS. This is reflected in the added sentence: “This trend is further supported by recent evidence showing that careful selection among pre-trained models can itself yield substantial gains in OOD generalisation and calibration performance (Naganuma et al., 2025).”
> > > > >
> > > > > We also refined the subsequent discussion of scaling limitations, now stating: “Notably, these models can still fail when pre-training data are poorly aligned with downstream domains, underscoring their sensitivity to data–domain mismatch and clarifying that scaling alone is not a universal solution for domain generalisation or drift resilience (Teterwak et al., 2025).”
> > > > >
> > > > > Together, these additions present the balanced dialectic the reviewer requested: empirical support for the strengths of scaling on standard benchmarks, paired with clear acknowledgement of its dependence on data–domain alignment.

---

> > > > > > ### Comment · Reviewer_6Sbw · 2025-11-27
> > > > > >
> > > > > > Thank you to the authors for their response.
> > > > > > I would just like to point out that the citation is not correctly reflected in the PDF.
> > > > > > Overall, my concerns have been resolved.

---

> > > > > > > ### Author Response · Authors · 2025-12-01
> > > > > > > **Response to Reviewer 6Sbw**
> > > > > > >
> > > > > > > Thank you for noticing. The (latexdiff-related) issue has been resolved in the current revision.

---

> ### Comment · Action_Editor_avza · 2025-12-08
> **Final recommendation**
>
> Dear reviewer,
>
> please consider authors feedback to all the raised concerns and provide a final recommendation for this work.
>
> Best,
> your Action Editor.

---

### Review · Reviewer_zqfA · 2025-10-08

**Summary Of Contributions:**

This paper provides a survey on ML for medical imaging and two threats to the trustworthiness of models: bias and data drift.
First, each of the two characteristics is explained and key concepts are outlined.
Afterwards, relevant works are discussed which address them in the medical domain.


## Strengths
- Relevant and important field of study
- Well written and comprehensible
- Clear examples for important aspects

## Weaknesses
- Novelty with regards to surveys focussing on one aspect
- Interaction of data drift and bias not discussed beyond introduction
- Limited scope

**Audience:**

Yes

**Audience Explanation:**

This study would be interesting for a part of the audience of TMLR.
It covers to interesting threats to the use of ML models in general (bias and data drift).
The main audience would be people interested in both, ML and the medical domain.
However, the writing style makes the content accessible to a wider audience.

**Broader Impact Concerns:**

no concerns

**Claims And Evidence:**

No

**Claims Explanation:**

This study provides accurate descriptions on the background of bias and data drift, as well as the collected studies.
However, I do not believe enough information is provided to show how this survey provides comprehensive information beyond the existing works.
Therefore the two main concerns are the novelty of the provided information and the comprehensiveness of collected publications:

# Novelty with regards to existing surveys
As stated in the introduction, there are surveys that address bias and data drift separately and the novelty lies in considering them together.
However, beyond one example in the introduction, the following section treat both concepts separate from each other.
This raises the question whether the content provided is different from the existing surveys.
This should be clarified, to show how the sections differ from information that is available in existing surveys.

More emphasis should be put on the interaction between bias and data drift, to show how existing works handle these two aspects, as this is the main novelty of this survey.

# Comprehensiveness and Scope
The number of collected works is comparatively small for a comprehensive survey. Table 1 lists 17 publications, two of which are not concerned with medical imaging. Moreover, non-demographic confounders are considered (Section 4), while the definition of bias in Section 2 states that it is interested in demographic confounders.
In Section 4.3, the last paragraph describes a bias mitigation method which does not fit well with the other works on synthetic dataset creation.
Furthermore, Section 5 only starts with data drift publications for the medical domain in Section 5.3. It is not clear why works outlined in Section 5.1 and 5.2 are relevant for the scope of this survey.
Lastly, there are no works included from 2025.

**Requested Changes:**

# Requested changes
- Discuss differences to existing surveys for the different sections
- Outline the methodology for paper collection and inclusion, or provide more support for the comprehensiveness of the collected papers
- Discuss intersection/combination of bias and data drift with relevant publications in more detail


# To strengthen work
- Clarify scope and remove works that do not fit (e.g., works not for medical images, non-demographic confounders)
- Consistent tense when describing existing works
- Check bracket use when citing (Section 3 has some cases where brackets should be removed)

---

> ### Author Response · Authors · 2025-11-25
> **Answer zqfA**
>
> We sincerely thank the reviewer for the thoughtful and constructive feedback, as well as for providing clear directions on how to strengthen our survey. Your comments were highly insightful and contributed meaningfully to improving both the clarity and scientific value of our work. We have carefully considered all points and applied the requested changes. In summary, the main concerns relate to the novelty and comprehensiveness of the survey, the integration of bias and data drift beyond the introductory section, the scope and selection of included works, and several consistency issues in presentation and referencing.
>
> ## Regarding the differences to existing surveys:
> The key distinction is that this survey unifies pre-deployment bias encoding assessment with post-deployment data-drift detection and label-free performance estimation, offering a lifecycle perspective that none of the existing surveys cover. Whereas prior surveys treat domain adaptation, OOD detection, or fairness and bias mitigation as separate, non-interacting topics, our work connects them within a single medical-imaging-specific reliability framework that reflects real clinical deployment needs.
> We have added a statement that clarifies this to the paper.
>
> ## Regarding the methodology for paper collection and the perceived limited scope:
> Following your suggestion, we added a short description of our paper collection process. Table 1 remains focused on the core contributions due to space constraints, but the main text now discusses a broader set of relevant works. We already clarified in the introduction why both demographic and non-demographic confounders are included.
>
> ## Regarding the intersection of bias and data drift:
> We thank the reviewer for this helpful suggestion. To emphasise the interaction between bias and data drift, we have revised Section 6 and introduced Table 2, which maps lifecycle stages to mitigation strategies while distinguishing bias- and drift-focused interventions. The updated text highlights how existing methods address both aspects jointly: pre-deployment data curation reduces subgroup imbalance and susceptibility to covariate drift; monitoring and drift-detection pipelines capture subgroup and temporal shifts; and domain adaptation or continuous retraining adjusts models while preserving subgroup fairness. These revisions clarify the survey’s core contribution by demonstrating how bias and drift interact across the ML lifecycle.
>
> ## Regarding clarification of scope and inclusion of works outside medical imaging:
> We now define bias more precisely and state explicitly that our survey focuses on performance disparities across subgroups defined by sensitive attributes, while discussing non-demographic confounders only when they contribute to such disparities. In addition, we clarify why non-medical drift-detection methods are included in early sections: these works are either foundational or directly inform the medical-imaging-specific approaches discussed later, and their inclusion provides essential context.
> We thank the reviewer for pointing out the tense and citation-bracket inconsistencies; we have corrected both throughout the manuscript.

---

> > ### Comment · Reviewer_zqfA · 2025-12-01
> >
> > Thank you for the updates and clarifications.
> > The new sections, in particular 6.1, are helpful.
> > While Section 3.4 is interesting and good for establishing the interaction of the two concepts, it is important to ensure its correctness.
> > This does not mean the introduced ideas are not, however there are not references for their support (e.g. equation 7).
> > For instance, there are a lot of different types of drifts and biases for which the equation needs to hold, and they theoretically do not need to be additive.
> > Also, I struggle a bit to understand the examples and their explanation.
> >
> > My other concerns remain to a large extent.
> > The novelty with existing surveys is only established for a small part of the content (e.g., Section 3.4).
> > Largely, this survey treats bias and data drift independently.
> > As such, Section 2-5 might have already been covered by existing works.
> > Potentially even parts of Section 6, as there only seems to be a single work that considered both concepts (Table 2).
> >
> > Scope concerns are unchanged. The introduction adds a sentence for the search procedure, but this is lacks detail to understand and follow how the search was conducted. Dates, keywords and venues are not mentioned and further details on the selection and filtering are missing.
> >
> > I agree that the works outside the medical scope provide important knowledge, however do not need not be included in Table 1 (in my opinion).

---

> > > ### Author Response · Authors · 2025-12-03
> > > **Answer zqFa**
> > >
> > > We thank the reviewer for the constructive feedback regarding the temporal decomposition. We agree that expressing this discrepancy as an additive decomposition may be an oversimplification, and it is not required for the conceptual point we wish to make. In the revised version, we therefore removed the additive formulation and strengthened the textual explanation. We also simplified the accompanying examples to illustrate the intersectionality between bias and data drift clearer and to avoid confusion. In particular, we emphasised the causal relationships of the examples and how biases and drifts can have similar causal relationships.
> > >
> > > Furthermore, we agree that purely non-medical works should not be included in Table 1. We have removed the non-medical works and streamlined the table accordingly. Similarly, we updated Table 2 by removing non-medical references such as DomainBed and IRM, while retaining GroupDRO as utilised in Alloula et al. (2025), and contextualising WILDS within Camelyon17-WILDS.
> > >
> > > Regarding the novelty aspect, we respectfully note that the value of a survey lies not only in introducing entirely new concepts, but in synthesizing scattered knowledge into a coherent framework. As acknowledged by the reviewer, Section 3.4's integration of bias and drift represents a novel contribution. The independent treatment in Sections 2-5 is intentional, to establish the necessary foundation before their synthesis. The reviewer's observation that "there only seems to be a single work that considered both concepts" (Table 2) precisely underscores the novelty and necessity of our integrated perspective.
> > >
> > > Alloula, A., Jones, C., Glocker, B., & Papież, B. W. (2025). Subgroups Matter for Robust Bias Mitigation. ICML.

---

> ### Comment · Action_Editor_avza · 2025-12-08
> **Final recommendation**
>
> Dear reviewer,
>
> please consider authors feedback to all the raised concerns and provide a final recommendation for this work.
>
> Best,
> your Action Editor.

---

### Review · Reviewer_bBiK · 2025-11-02

**Summary Of Contributions:**

The paper „Bias Assessment and Data Drift Detection in Medical Image Analysis: A Survey” provides an overview of biases and data drift within the medical domain. The paper provides a systematic overview of the different types and sources of both biases and data drift. For each type, the paper provides examples from the literature and discusses how the biases/data drift can be detected.

Strengths:
The paper is very well-written and well-structured and easy to read. Indeed, for people not aware or with only basic knowledge about biases and data drift, this paper is a very good starting point to understand both concepts very well. This goal is also stated within the paper, and I can say that the authors clearly succeeded. The illustrative examples are well-placed within the paper and make both the causal constructs clearer and aid with the understanding of more complex measurement concepts.

Weaknesses:
I lack a scientific insight from this paper. That biases and data drift can be considered together is not sufficient for this. Indeed, my very positive comments above can also be phrased negatively from another perspective: This is a very good book chapter on the topic, but not a journal paper providing new scientific insights.

For such insights, I miss two things: a clear description of how the literature search for the review was conducted that allows me to understand how complete the discussion of the literature is (e.g., search terms, inclusion/exclusion criteria, etc.) and then some synthesis of the results, e.g., by pointing out missing pieces, contradictions in prior work, or how prior works supports each other.

**Audience:**

Yes

**Audience Explanation:**

Starters on the discussed topic will find the paper helpful. People already familiar with the concepts will likely not learn anything new.

**Claims And Evidence:**

Yes

**Claims Explanation:**

All claims in the paper are supported. However, this statement needs to be seen with a grain of salt, as the review is currently only illustrative and a systematic search is missing. Given the lack of such a search (and also no indication that this was attempted or desired), everything is clear, though it is unclear how complete the results are.

**Requested Changes:**

Clarify how the paper advances the state-of-the-art. Note, that simply stating that bias and drift should be discussed together, without a clear, evidence-based or theoretically derived justification is not sufficient.

---

> ### Author Response · Authors · 2025-11-25
> **Answer bBiK**
>
> We thank the reviewer for the insightful feedback. In summary, the concern is that while the survey is clearly written, it currently lacks the scientific depth expected of a journal article, particularly in terms of articulating a clear scientific contribution, and offering a stronger discussion of prior work. We address all these issues by providing a clear structure in the form of an improved roadmap and highlighting key findings in our manuscript.
>
> ## Regarding clarification of the contribution and advancement of the state-of-the-art:
> We thank the reviewer for this important comment. To clarify how our paper advances the state of the art, we have revised Section 6 and the Conclusion to explicitly demonstrate why bias and data drift should be considered together. We provide an evidence-based justification through the crosswalk in Table 2, which maps lifecycle stages to mitigation strategies and distinguishes bias- and drift-focused interventions. This mapping shows, with supporting references from medical imaging, how existing methods handle bias, drift, or both jointly, highlighting gaps where integration is needed. By linking pre-deployment and post-deployment strategies and showing their interaction across the ML lifecycle, the survey provides a theoretically grounded and practical framework for understanding and mitigating bias-drift interactions, thus advancing the field beyond prior work that treats these aspects separately.
>
> Therefore, our paper does not merely state that these aspects should be discussed together, but offers a structured, evidence-based rationale for why their integration is necessary and how it can be operationalized in practice.
>
> ## Regarding the lack of scientific insight:
> To provide a clear scientific insight while remaining true to the nature of a survey paper, we have revised the conclusion to include a more explicit take-home message. The updated text highlights that biases and data drift are interrelated challenges in medical AI, and that only by jointly monitoring and managing these threats can researchers and practitioners develop systems that are fair, robust, and clinically reliable. This revision clarifies the key insights of the survey and provides readers with actionable guidance grounded in the current state of the field.
>
> ## Regarding the missing description of the literature search:
> Thank you for pointing this out. The manuscript did not describe how the literature was collected. Our initial search was not based on a predefined protocol but relied on keyword queries and screening recent publications in major venues. We now state this explicitly in the manuscript to clarify how the included literature was identified.
>
> ## Regarding the lack of connection across prior work:
> We have expanded the discussion section to better articulate the synthesis of how prior methods relate to each other. Specifically, we highlight how most prior work is complementary to each other. We agree that these additions improve the integrative value of the survey.

---

> > ### Comment · Reviewer_bBiK · 2025-12-03
> >
> > Unfortunately, my main criticism is unresolved.
> >
> > For a survey paper, the analysis of the literature is not sufficiently systematic. The link between pre-deployment and post-deployment strategies might be something that is articulated cleanly here. However, for me this is not sufficient in terms of scientific depth, as the existence of such a link is fairly obvious. Since there is no theoretically support or empirical data that quantifies how important this link is and what we loose (or not loose) when not considering this link, the value remains limited.

---

> > > ### Author Response · Authors · 2025-12-08
> > > **Response on Systematic Analysis of the DG–DA Link**
> > >
> > > We thank the reviewer for this comment and agree that simply stating the existence of a link between pre-deployment and post-deployment strategies would not, on its own, constitute sufficient scientific depth. To address this, we have added a dedicated subsection (Sec. 6.2) that presents a systematic analysis of studies explicitly linking design-time generalisation (DG) and post-deployment adjustment (DA). For each included work, we extract the targeted lifecycle stage (Tab. 2) and the specific methodological mechanism through which DG and DA are connected.
> > >
> > > This systematic review demonstrates that, despite being conceptually intuitive, explicit DG-DA pipelines remain limited and fragmented, with most methods addressing only a single lifecycle stage in isolation. We also acknowledge the reviewer’s point that the literature currently lacks strong theoretical or quantitative evidence quantifying the benefit of jointly considering DG and DA. Rather than claiming such evidence, our analysis identifies this absence as a central research gap.
> > >
> > > By making these gaps explicit and highlighting underexplored DG-DA combinations, the section provides a structured diagnosis of current limitations and a clear motivation for future lifecycle-aware theoretical and empirical studies, rather than merely asserting an obvious connection.

---

### Author Response · Authors · 2025-11-25
**Rebuttal for Bias Assessment and Data Drift Detection in Medical Image Analysis: A Survey**

We sincerely thank the reviewers for their time and effort in assessing our manuscript and for the constructive and encouraging feedback. We appreciate the recognition that the paper is well-written, well-structured, and easy to read (bBiK, zqfA), and we are pleased that it is regarded as addressing an important field of study (zqfA). We also value the feedback that the exposition is helpful for readers with limited prior knowledge of biases and data drift to understand both concepts very well (bBiK), and that it translates complex fairness criteria and drift types into clinically interpretable terms in line with our goal of bridging communities (6Sbw). We thank the reviewers for noting that the illustrative examples are clear, well-placed, and supportive of understanding the underlying causal and measurement concepts (bBiK, zqfA). In addition, we appreciate the positive remarks regarding the practical lifecycle framing and visual taxonomy linking bias and drift, which supports clinicians in reasoning about reliability checkpoints (6Sbw), as well as the breadth of methodological coverage and the grounding of these methods in medical-imaging use cases (6Sbw).

To address the individual concerns, we carefully considered all comments and applied changes to the paper where appropriate. We have uploaded a revised manuscript that highlights all modifications and added a list of all changes to the rebuttal. An exhaustive list of changes is given below.

In summary, we refined the manuscript’s structure, added clearer definitions and explanations, and substantially expanded the discussion of mitigation strategies. We also introduced a new crosswalk table that maps lifecycle stages to robustness and drift-related methods, together with additional clarifications on literature scope, bias–drift distinctions, and methodological interplay.

We welcome the opportunity for continued discussion to further improve the manuscript.


# Summary of Changes in the Revised Manuscript
We have marked all revisions directly in the updated manuscript. The key changes are outlined below.

## General Manuscript Improvements
- Clarified the final sentence of the conclusion to better convey the main insight and take-home message.
- Added a short description of the literature search and paper collection procedure to allow readers to assess the scope and coverage of the reviewed work.
- Improved the definition and scope of bias in Section 2.
- Added clarifications regarding the inclusion of non-medical drift detection methods in Sections 3, 5.1, and 5.2.
- Fixed several presentation inconsistencies throughout the manuscript.

## Structural Revisions
- Updated the paper roadmap at the end of the introduction to more clearly describe the types of biases and data drift covered.
- Added a dedicated subsection (6.1) devoted to “Mitigation Strategies”.

## New Content and Expanded Explanations
- Added a new table (Table 2): “Crosswalk mapping lifecycle stages to Design-time Robustness (DG) and Post-deployment Adjustment (DA)”, summarizing how robustness and drift-related methods align with different stages of the ML lifecycle.
- Added additional mitigation strategies in Section 6, including IRM, Group-DRO, resampling, and related approaches.
- Added a new subsection (Section 3.4) introducing a formal time-dependent decomposition of bias vs. drift.
- Included a causal and anticausal perspective with illustrative examples.
- Expanded the discussion on how mitigation methods may support or contradict one another depending on the clinical deployment context.

## Reorganization of Methods
- Moved domain adaptation (DA) and OOD detection methods from the previous Section 6 into the updated Section 6.1 on mitigation strategies.

---

### Decision · Action_Editor_avza · 2025-12-15

**Recommendation:** Reject

**Audience:**

Yes

**Audience Explanation:**

While I believe that the presented survey may be of interest to a specific audience, it could be substantially improved considering the criticism brought by the reviewers. In particular, I believe that bias and data drift are a mainstay in critical decision-making systems. Nevertheless, a deeper analysis of existing approaches might further strengthen this work, resulting in a greater interest to the community.

**Claims And Evidence:**

No

**Claims Explanation:**

The presented manuscript is well written and organized, and offers some interesting perspectives (e.g., linking bias and data drift). Nevertheless, the three reviewers agree that the actual contribution is merely organizational, lacking comprehensiveness and conceptual depth. Reviewers also note that the proposed unifying aspect is rather weak, with substantial material potentially overlapping with existing surveys in closely-related topics.

**Resubmission Of Major Revision:**

The authors may consider submitting a major revision at a later time.